# FAIRNESS-ENHANCING MIXED EFFECTS DEEP LEARNING IMPROVES FAIRNESS ON IN- AND OUT-OF-DISTRIBUTION CLUSTERED (NON-IID) DATA

## ABSTRACT

Traditional deep learning (DL) suffers from two core problems. Firstly, it assumes training samples are independent and identically distributed. However, numerous real-world datasets group samples by shared measurements (e.g., study participants or cells), violating this assumption. In these scenarios, DL can show compromised performance, limited generalization, and interpretability issues, coupled with cluster confounding causing Type 1 and 2 errors. Secondly, models are typically trained for overall accuracy, often neglecting underrepresented groups and introducing biases in crucial areas like loan approvals or determining health insurance rates, such biases can significantly impact one's quality of life. To address both of these challenges simultaneously, we present a mixed effects deep learning (MEDL) framework. MEDL separately quantifies cluster-invariant fixed effects (FE) and cluster-specific random effects (RE) through the introduction of 1) a cluster adversary which encourages the learning of cluster-invariant FE, 2) a Bayesian neural network which quantifies the RE, and a mixing function combining the FE an RE into a mixed-effect prediction. We marry this MEDL with adversarial debiasing, which promotes equalized odds fairness across FE, RE, and ME predictions for fairness-sensitive variables. We evaluated our approach using three datasets: two from census/finance focusing on income classification and one from healthcare predicting hospitalization duration, a regression task. Our framework notably enhances fairness across all sensitive variables-increasing fairness up to 73% for age, 47% for race, 83% for sex, and 26% for marital-status. Besides promoting fairness, our method maintains the robust performance and clarity of MEDL. It's versatile, and suitable for various dataset types and tasks, making it broadly applicable. Our GitHub repository publicly houses the implementation.

## 1 INTRODUCTION AND RELATED WORKS

Deep learning (DL) has become an integral tool in many applications, such as computer vision Krizhevsky et al. (2012); Nguyen & Manry (2021), finance Shi et al. (2022), and healthcare Nguyen et al. (2020). However, its application to real-world datasets reveals inherent limitations, notably in handling clustered samples where measurements are drawn from the same entities. This clustering often leads to reduced prediction performance and interpretability, along with biases that disproportionately affect minority groups in areas like finance and healthcare. This is especially harmful because 1) an unfair model can produce inaccurate predictions impacting life-altering decisions, 2) unfairness towards a particular group can lead to those individuals receiving poorer care or inadequate access to loans, and 3) an unfair model perpetuates inequities present in society. Navarro et al. (2021) described fairness issues in machine learning across medical specialties, highlighting the compromises often made for minority groups. Despite these inequities, an integrated DL framework handling sample clustering and enhancing fairness has remained unsolved. This work introduces a robust fairness-enhancing mixed effects deep learning (MEDL) framework that retains the benefits of a top-performing ARMED approach for mixed effects deep learning Nguyen et al. (2023) that we recently developed, while simultaneously enhancing fairness. This MEDL framework enhances prediction accuracy on clustered data, increases model interpretability by separately predicting cluster-invariant FE and cluster-dependent ME, and reduces biases towards subgroups.

Several MEDL methods have been introduced as potential solutions to the aforementioned clustering including MeNets Xiong et al. (2019) and LMMNN Simchoni & Rosset (2021), as well as ARMED Nguyen et al. (2023), with ARMED comparing favorably across multiple problem domains. Meanwhile, the causes of unfairness in machine learning (ML) Chouldechova & Roth (2020) can be categorized into those that stem from biases in the data and those that arise from biases in the ML algorithm. This work focuses on alleviating algorithmic biases. Methods for addressing algorithmic unfairness typically fall into one of 3 categories: pre-process, in-process, post-process. We choose an in-process approach because it imposes the required accuracy-to-fairness trade-off in the objective function Woodworth et al. (2017). Meanwhile, pre-process methods can harm the explainability of the results and leave high uncertainty regarding accuracy at the end of the process Pessach & Shmueli (2022), and post-process methods require the final decision maker to possess the sensitive group membership information, which may not be available at that time Woodworth et al. (2017). Among in-process methods, Yang et al. (2023) used adversarial debiasing with a traditional DL multilayer perceptron. They attained an improvement in fairness while classifying COVID-19 severity (high vs. low); however, the authors did not assess the statistical significance of the improvement. Beutel et al. (2019) introduced an alternative to adversarial debiasing, called absolute correlation loss, which acts as a regularization term for the loss function and can significantly improve equity of the false positive rate. Neither works, however, combined fairness-enhancement with a MEDL model to separate and properly quantify FE and RE. The profound lack of ML research at the intersection of mixed effects modeling and fairness-enhancement motivates our work.

This work makes the following contributions: **1.** We introduce a comprehensive, general framework that simultaneously addresses cluster variance and enhances fairness for sensitive variables, setting the stage for fairer and more reliable machine learning results. **2.** We rigorously test our method using a diverse set of real-world datasets from finance and medicine, covering both classification and regression tasks. **3.** Our results demonstrate statistically significance, large improvements in fairness across all datasets, tasks, and fairness sensitive variables: race, age, sex, and marital-status.

## 2 METHODS

### 2.1 MEASURING FAIRNESS THROUGH EQUALIZED ODDS

Equalized odds Hardt et al. (2016) states that a predictive model is fair if its prediction, $\hat{y}$ and the sample's sensitive group membership, $s$, are conditionally independent given its true label $y$. This metric was chosen as the metric of fairness because 1) it readily extends to classification and regression Zhang et al. (2018), 2) it can be easily implemented in neural networks as an in-process method, and 3) it can handle sensitive variables with $>2$ categories (e.g., race) Yang et al. (2023). In classification tasks, this metric can be assessed by measuring the standard deviation of both the true positive rate (TPR) and false positive rate (FPR) across categories within each sensitive variable, with maximal fairness achieved when the standard deviation is 0. For regression, the standard deviation of mean squared error (MSE) values is used, rather than TPR and FPR.

### 2.2 CONSTRUCT A FAIRNESS-ENHANCING, MIXED EFFECT DEEP LEARNING FRAMEWORK

#### 2.2.1 BEGIN WITH THE TOP PERFORMING MEDL FRAMEWORK, ARMED.

Nguyen et al. (2023) introduced the Adversarially-Regularized Mixed Effect Deep Learning (ARMED) framework, demonstrating unparalleled ability to improve performance on in-distribution data, improve generalization for out-of-distribution (OOD) data, improve interpretability, and mitigate confounding. Further, ARMED is agnostic to neural network architecture. Beginning with a base neural network for prediction—a feedforward neural network is shown for illustration as subnetwork 1, in Fig. 1—the ARMED framework adds several novel components. 1) An Adversarial Classifier, $A_c$, (Subnetwork 2 in the fig.) is added to encourage the base network to learn features *not* predictive of the sample's associated cluster, thereby teaching the network to output cluster-invariant Fixed Effects (FE). 2) A Random Effects (RE) Bayesian subnetwork (subnetwork 4 in the fig.) is added to learn to predict cluster-specific slopes and intercepts denoted as RE, and 3) a mixing function is added to combine the FE and RE into the cluster-specific mixed effects (ME) predictions. To use the model to predict on data from clusters unseen during training (OOD data), a separate cluster membership network is trained (network 6 in the fig.).

The objective function for ARMED is:

$$L_e(y, \hat{y}_M) + \lambda_F L_e(y, \hat{y}_F) - \lambda_g L_{CCE}(Z, \hat{Z}) + \lambda_K D_{KL}(q(U) \parallel p(U)) \tag{1}$$

Where $L_e$ represents the cross-entropy loss. $y$, $\hat{y}_M$, and $\hat{y}_F$ signify the ground truth label, Mixed Effects prediction, and Fixed Effects prediction, respectively. $L_{CCE}$ denotes the categorical cross-entropy loss of the Adversarial Classifier (2) predicting the cluster. $Z$ and $\hat{Z}$ are the cluster ID and its prediction. The term $D_{KL}(q(U) \parallel p(U))$ is the KL-divergence between the Bayesian neural network submodel's approximation, $q(U)$, and the prior posterior, $p(U)$. The hyperparameters $\lambda_F$, $\lambda_g$, and $\lambda_K$ are associated with $L_e$, $L_{CCE}$, and $D_{KL}$, respectively, and their values are tuned using hyperparameter optimization. While ARMED has shown promise in enhancing prediction accuracy for non-iid data, it lacks fairness constraints and was restricted to classification tasks, limitations we alleviate in this work.

### 2.2.2 FIRST, CONSTRUCT A FAIRNESS-ENHANCING DOMAIN ADVERSARIAL FRAMEWORK (SUBSET OF FULL FRAMEWORK)

As in the ARMED testing in Nguyen et al. (2023), we form a subset of the ARMED model, known as the domain adversarial (DA) model. This simpler model consists of ARMED without the RE subnetwork (i.e., it includes subnetworks 1,2, and subnetwork 4 is left out, see Fig. 1). This facilitates rapid experimentation —in this work, the introduction of fairness-enhancement. For the classification tasks, the DA model was built with a loss function:

$$\lambda_y L_{BCE}(y_F, \hat{y}_F) - \lambda_z L_{CCE}(Z, \hat{Z}) \tag{2}$$

implementing binary cross-entropy and categorical cross-entropy losses for the FE classifier and the cluster adversary, $A_C$, respectively as well as hyperparameters for the FE classifier ($\lambda_y$) and cluster adversary ($\lambda_z$). For regression tasks, the main regressor uses MSE loss.

The domain adversarial (DA) model is made fairness-enhancing by adding a fairness-promoting adversary for the fixed effects captured by the DA model. Because we are adding adversarial debiasing to the DA model we denote it as fair DA adv. deb.; it utilizes the adversarial subnetwork $A_F$ (subnetwork 3 in Fig. 1). The subnetwork attempts to predict a sample's sensitive features based on the main predictor's estimated target, $\hat{y}_F$, and the true target $y$. It was added to the base domain adversarial model with the following loss objective:

$$\lambda_y L_{BCE}(y, \hat{y}_F) - \lambda_z L_{CCE}(Z_c, \hat{Z}_c) - \lambda_s L_{CCE}(S, \hat{S}) \tag{3}$$

with an additional hyperparameter ($\lambda_s$) for the adversarial debiasing subnetwork allowing for tuning of fairness through hyperparameter optimization, e.g., BOHB Falkner et al. (2018). In our experiments, we compare the above two models (DA, fair DA adv. deb.), to the base neural network (subnetwork 1 in Fig. 1) and to another approach for fairness, known as absolute correlation loss. Introduced by Beutel et al. (2019), in lieu of adding an adversary, this approach adds a loss term to promote fairness. We denote this model by fair DA ACL. A Pearson correlation coefficient is computed between the predictor output and sensitive subgroups. A correlation loss term can be added for each sensitive feature, incentivizing learning that reduces overall correlation according to the following total loss function:

$$\lambda_y L_{BCE}(y_F, \hat{y}_F) - \lambda_z L_{CCE}(Z, \hat{Z}) + \lambda_s(|L_{corr}(\hat{y}_F, S_1)| + ... + |L_{corr}(\hat{y}_F, S_n)|) \tag{4}$$

for sensitive features $S_1$ through $S_n$ and with an added correlation loss weight hyperparameter ($\lambda_s$).

### 2.2.3 CONSTRUCT THE PROPOSED FULL, FAIRNESS-ENHANCING ARMED FRAMEWORK

Our proposed full fairness-enhancing ARMED framework introduces both additional adversarial debiasing subnetworks $A_F$ and $A_M$ (subnetworks 3 and 5 in Fig. 1) to the full ARMED model. These subnetworks encourage fairness in the FE and ME predictions. Their incorporation involves new loss terms, yielding the following overall loss function:

$$L_e(y, \hat{y}_M) + \lambda_F L_e(y, \hat{y}_F) - \lambda_g L_{CCE}(Z, \hat{Z}) + \lambda_K D_{KL}(q(U) \parallel p(U)) - \lambda_D L_{CCE}(S, \hat{S}) \tag{5}$$

Here, $S$ and $\hat{S}$ correspond to the sensitive features and their predictions within the debiasing adversarial network. The weight $\lambda_D$ modulates the loss of the adversarial debiasing subnetworks. These hyperparameters can also be readily tuned using BOHB Falkner et al. (2018).

## 2.3 MATERIALS

We tested our models 3 datasets: Adult, IPUMS, and Heritage Health spanning different sectors (finance, health care) and tasks (classification and regression). *The Adult dataset* Becker & Kohavi (1996) derived from US Census data, contains 14 demographic and employment attributes. Labels indicate whether or not a person earns >$50,000 a year. The random effect, *occupation*, captures a sample of 15 of the possible occupations and induces clustered sample correlation as similar occupations provide comparable salaries. Fairness-sensitive variables are *sex* (male, female), *age* bracket (<30, 31-45, >45), *race* (black, white, American Indian/Alaska Native, Asian, other), and *marital-status* (6 categories). After data cleaning, the dataset contains 23,002 samples (people).

The *IPUMS dataset* Ruggles et al. (2023) was sourced from the 2006 American Community Survey census. This dataset includes records from 1,572,387 US residents, covers 14 socio-demographic attributes, and specifies whether an individual's income is >$50,000. The samples are clustered by occupation (i.e., the random effects) for which there are 15 different categories. Fairness-sensitive variables mirror those in the Adult dataset (*Sex, Age, Race, Marital-status*), but *Race* is expanded to include additional classes (Chinese, Japanese, other API, 2 major races, 3+ major races). This dataset was used to test the scalability of our proposed fairness-enhancing ARMED framework on a dataset 60x larger than the Adult dataset.

To test how well the proposed framework would generalize to data from a different sector (healthcare rather than finance) and regression rather than classification, we used the *Heritage Health dataset*. This dataset contains healthcare insurance claims for patients affiliated with the California-based Heritage Provider Network. These records were aggregated into a yearly summary of all claims per patient, comprising 19 features (e.g., age, number of claims, place of service, specialty). These summary records cover 76,013 patients. The target is the number of days spent in the hospital in the following year. The sensitive variables are *age* bracket (0-29, 30-49, 50-69, 70-79, 80+) and *sex* (male, female, NA), while the primary healthcare provider (2708 total providers) is the random effect.

## 2.4 DATA PARTITIONING AND PERFORMANCE EVALUATION

For model training, each dataset was partitioned into *seen* and *unseen* subsets based on the random effect. In the Adult dataset, the 6 highest-frequency occupations (71% of samples) constituted seen data, for IPUMS, the top 5 occupations (70%), and for Heritage Health, the top 50 providers (50%) were used. Data from the seen clusters were used for model training, and the remaining data was held-out as unseen clusters to test generalizability. Seen data was further partitioned via 10-fold cross-validation with 8 folds to train a model, one fold for validation, and one fold for test (i.e., which was used for all reported seen cluster performance metrics). The test metrics are compiled across all 10 folds. Depending on the task (classification or regression) performance metrics are balanced accuracy or MSE and fairness is evaluated as the standard deviation of the TPR and FPR or the std dev of the MSE across categories within each sensitive variable.

## 3 RESULTS

### 3.1 ADULT DATASET

Table 1 indicates that while both Domain Adversarial Debiased (fair DA adv. deb.) and Domain Adversarial with absolute correlation (fair DA ACL) enhance fairness, fair DA adv. deb. exhibits a more consistent fairness improvement across all sensitive variables. Moreover, fair DA adv. deb. enhances fairness with minimal reduction in balanced accuracy compared with fair DA ACL—1% vs 1.6%. *Given these findings, we chose to incorporate fair DA adv. deb. into the ARMED framework to bolster its fairness capabilities.*

From the results delineated in Table 2, the efficacy of the fairness-enhancing ARMED framework on the Adult data is evident. On occupations seen during training, the ARMED ME baseline is slightly fairer than the ARMED FE baseline on both TPR and FPR. There are huge improvements when the debiasing heads are added. For the sensitive variable *Age*, the model's FE predictions TPR standard deviation was reduced from 0.173 to 0.092, a 46.8% reduction. Simultaneously, its FPR standard deviation showed a 20.5% reduction, decreasing from 0.151 to 0.12. Enhanced fairness is observed

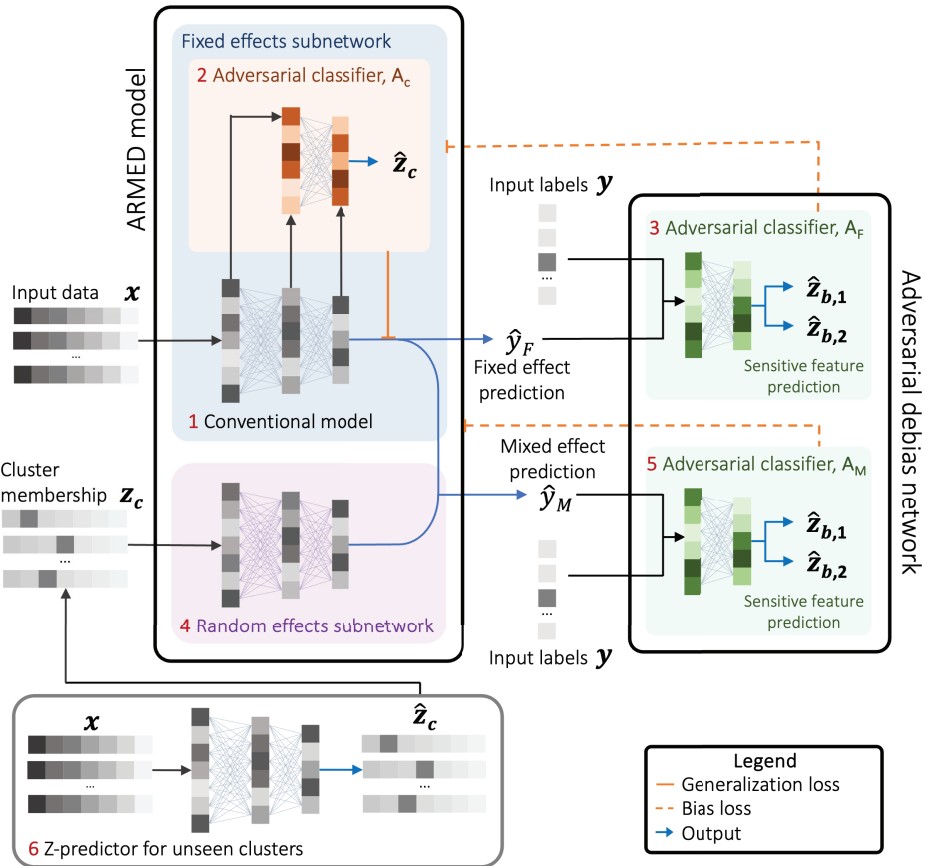

Figure 1: Fairness enhancing ARMED framework with adversarial debiasing for both fixed and mixed effects.

across the other sensitive variables. For *Marital-status*, the TPR standard deviation dropped 15% (from 0.206 to 0.175). For *Race*, the reduction in TPR was 24.8%, and for *Sex*, it was 70.5%. These marked reductions underscore the consistent improvement in fairness that the fairness-enhancing ARMED framework attains for the FEs. Similar improvements are observed for the ME predictions. Regarding prediction accuracy, the fair ARMED FE predictions achieved a balanced accuracy of 80% for *Age*, which is a modest drop of 0.8% from the non-fairness counterpart. This negligible compromise in accuracy is a testament to the model's ability to balance fairness without significantly sacrificing performance.

For the clusters unseen during training, the fairness-enhancing ARMED framework continues to perform well. For *Age*, there was a 54.5% reduction in TPR standard deviation in the FE predictions compared to its non-fairness counterpart. For the sensitive variable *Sex* in the FE predictions, the TPR standard deviation reduced by 66.7%, demonstrating the model's ability to promote fairness in OOD data.

## 3.2 IPUMS DATASET

The results shown in Table 3 show that the fairness-enhancing ARMED framework scales well to the much larger IPUMS dataset and demonstrates a substantial improvement in prediction fairness. In particular, the standard deviations of TPR and FPR within each fairness-sensitive variable's categories show a marked reduction compared to the standard ARMED framework. **In this larger dataset, we observe this improvement for every single fairness-sensitive variable.** For example, our results on the test folds from the clusters (occupations) seen during training, show the TPR standard deviation for *Age* decreased from 0.161 to 0.059 in fair ARMED FE predictions, a 63.3% improvement in fairness. Similarly, for the *Race*, the TPR standard deviation was reduced from

Table 1: Prediction of income level using the ADULT dataset with the Domain Adversarial models.

| Model | Sensitive feature | TPR standard deviation | | | FPR standard deviation | | | Balanced accuracy (%) | | |
|---|---|---|---|---|---|---|---|---|---|---|
| | | Mean | 95% CI | p-value* | Mean | 95% CI | p-value* | Mean | 95% CI | p-value* |
| | | Seen occupations | | | | | | | | |
| DFNN | Age | 0.163 | 0.155 - 0.170 | | 0.233 | 0.232 - 0.235 | | **0.809** | 0.807 - 0.811 | |
| | Marital-status | 0.187 | 0.175 - 0.202 | | 0.348 | 0.346 - 0.351 | | | | |
| | Race | 0.149 | 0.096 - 0.198 | | 0.129 | 0.121 - 0.136 | | | | |
| | Sex | 0.098 | 0.091 - 0.104 | | 0.211 | 0.210 - 0.212 | | | | |
| DA | Age | 0.169 | 0.156 - 0.177 | $p < 0.001$ | 0.234 | 0.232 - 0.235 | 0.01 | 0.808 | 0.806 - 0.811 | $p < 0.001$ |
| | Marital-status | 0.200 | 0.186 - 0.214 | $p < 0.001$ | 0.351 | 0.350 - 0.354 | $p < 0.001$ | | | |
| | Race | 0.155 | 0.109 - 0.199 | 0.47 | 0.130 | 0.124 - 0.135 | 0.33 | | | |
| | Sex | 0.102 | 0.092 - 0.111 | $p < 0.001$ | 0.212 | 0.210 - 0.214 | $p < 0.001$ | | | |
| fair DA ACL | Age | 0.166 | 0.154 - 0.174 | 0.09 | 0.228 | 0.225 - 0.231 | $p < 0.001$ | 0.802 | 0.774 - 0.806 | $p < 0.001$ |
| | Marital-status | **0.181** | 0.165 - 0.203 | 0.02 | **0.338** | 0.337 - 0.341 | $p < 0.001$ | | | |
| | Race | 0.133 | 0.091 - 0.185 | 0.04 | 0.129 | 0.123 - 0.135 | 0.89 | | | |
| | Sex | **0.058** | 0.052 - 0.065 | $p < 0.001$ | 0.199 | 0.197 - 0.202 | $p < 0.001$ | | | |
| fair DA adv. deb. | Age | **0.131** | 0.094 - 0.171 | $p < 0.001$ | **0.222** | 0.214 - 0.230 | $p < 0.001$ | 0.790 | 0.785 - 0.797 | $p < 0.001$ |
| | Marital-status | 0.184 | 0.156 - 0.205 | 0.33 | 0.345 | 0.338 - 0.351 | $p < 0.001$ | 0.805 | 0.802 - 0.808 | $p < 0.001$ |
| | Race | **0.106** | 0.072 - 0.150 | $p < 0.001$ | **0.119** | 0.113 - 0.126 | $p < 0.001$ | 0.807 | 0.805 - 0.810 | $p < 0.001$ |
| | Sex | 0.075 | 0.057 - 0.096 | $p < 0.001$ | **0.198** | 0.193 - 0.204 | $p < 0.001$ | 0.789 | 0.780 - 0.795 | $p < 0.001$ |
| | | Unseen occupations | | | | | | | | |
| DFNN | Age | 0.207 | 0.195 - 0.216 | | 0.103 | 0.101 - 0.105 | | **0.772** | 0.770 - 0.775 | |
| | Marital-status | 0.228 | 0.215 - 0.242 | | 0.123 | 0.121 - 0.125 | | | | |
| | Race | 0.151 | 0.108 - 0.190 | | 0.049 | 0.047 - 0.051 | | | | |
| | Sex | 0.067 | 0.059 - 0.076 | | 0.064 | 0.063 - 0.066 | | | | |
| DA | Age | 0.212 | 0.198 - 0.226 | 0.008 | 0.104 | 0.103 - 0.106 | 0.02 | 0.771 | 0.768 - 0.774 | 0.002 |
| | Marital-status | 0.231 | 0.221 - 0.242 | 0.11 | 0.124 | 0.123 - 0.126 | 0.001 | | | |
| | Race | 0.149 | 0.112 - 0.176 | 0.76 | 0.050 | 0.047 - 0.051 | 0.27 | | | |
| | Sex | 0.065 | 0.057 - 0.073 | 0.24 | 0.064 | 0.063 - 0.065 | 0.19 | | | |
| fair DA ACL | Age | 0.216 | 0.201 - 0.223 | $p < 0.001$ | **0.097** | 0.095 - 0.100 | $p < 0.001$ | 0.756 | 0.736 - 0.761 | $p < 0.001$ |
| | Marital-status | 0.233 | 0.216 - 0.253 | 0.06 | **0.114** | 0.112 - 0.118 | $p < 0.001$ | | | |
| | Race | **0.105** | 0.074 - 0.142 | $p < 0.001$ | 0.047 | 0.045 - 0.049 | $p < 0.001$ | | | |
| | Sex | **0.011** | 0.003 - 0.018 | $p < 0.001$ | **0.056** | 0.054 - 0.058 | $p < 0.001$ | | | |
| fair DA adv. deb. | Age | **0.159** | 0.099 - 0.222 | $p < 0.001$ | 0.101 | 0.094 - 0.105 | $p < 0.001$ | 0.760 | 0.751 - 0.768 | $p < 0.001$ |
| | Marital-status | **0.219** | 0.190 - 0.239 | $p < 0.001$ | 0.121 | 0.115 - 0.125 | $p < 0.001$ | 0.765 | 0.757 - 0.770 | $p < 0.001$ |
| | Race | 0.117 | 0.088 - 0.156 | $p < 0.001$ | **0.045** | 0.042 - 0.048 | $p < 0.001$ | 0.768 | 0.765 - 0.771 | $p < 0.001$ |
| | Sex | 0.023 | 0.002 - 0.050 | $p < 0.001$ | 0.060 | 0.056 - 0.065 | $p < 0.001$ | 0.755 | 0.744 - 0.761 | $p < 0.001$ |

*All p-values are measured against the baseline classifier (DFNN)*

*DFNN: dense feedforward neural network; DA: domain adversarial model; fair DA ACL: domain adv. model with absolute correlation loss; fair DA adv. deb.: domain adv. mdl with adversarial debiasing; CI: conf. interval. Metrics were computed through 30 trials of each mdl across 10 training folds with different random seeds. Binary classification threshold was set via the Youden pt. determined at each training fold. The best fairness results (lowest std. dev. of classification accuracy across categories within each sensitive variable) are bolded.*

0.134 to 0.077 in fair ARMED FE predictions, indicating a 43% improvement. Further, for the sensitive variable *Sex*, we observe a 35.7% enhancement in fairness, with TPR standard deviations dropping from 0.140 in ARMED FE to 0.09 in *fair* ARMED FE. A similar improvement in fairness for all variables is observed for the fair ARMED *ME* predictions. Moreover, these improvements are not limited to seen occupations (the in-distribution-data tests). On data from clusters (occupations) unseen during training (OOD data), we find similar enhancements. Across *Age* brackets, we observe an impressive 51.35% improvement in TPR fairness (from 0.261 in ARMED FE to 0.127 in fair ARMED FE). Across categories of *Race*, fairness improves by approximately 36.7%, with standard deviations dropping from 0.158 in ARMED FE to 0.1 in fair ARMED FE. Again, similarly, excellent results are observed for the fair ARMED ME predictions.

Importantly, these reductions in standard deviations for both TPR and FPR across various sensitive attributes are *statistically significant*, with p-values consistently less than 0.001 when compared against the baseline ARMED FE and ME. Meanwhile, balanced accuracy exhibits only minor fluctuations. We find this marginal trade-off acceptable given the substantial improvements to fairness.

## 3.3 HERITAGE HEALTH DATASET

The Heritage Health dataset results demonstrate the fairness-enhancing ARMED framework's efficacy in enhancing fairness in a regression task across each fairness-sensitive variable, *Age* and

Table 2: Prediction of income level using the ADULT dataset with two Debias head fairness enhancing ARMED framework.

| Model | Sensitive feature | TPR standard deviation | | | FPR standard deviation | | | Balanced accuracy (%) | | |
|---|---|---|---|---|---|---|---|---|---|---|
| | | Mean | 95% CI | p-value* | Mean | 95% CI | p-value* | Mean | 95% CI | p-value* |
| | | | | Seen occupations | | | | | | |
| ARMED FE baseline | Age | 0.173 | 0.157 - 0.189 | | 0.151 | 0.142 - 0.160 | | | | |
| | Marital-status | 0.206 | 0.173 - 0.255 | | 0.224 | 0.211 - 0.240 | | 0.808 | 0.806 - 0.810 | |
| | Race | 0.153 | 0.079 - 0.253 | | 0.076 | 0.065 - 0.092 | | | | |
| | Sex | 0.105 | 0.090 - 0.118 | | 0.142 | 0.131 - 0.154 | | | | |
| Fair ARMED FE | Age | **0.092** | 0.037 - 0.149 | $p < 0.001$ | **0.120** | 0.091 - 0.155 | $p < 0.001$ | 0.800 | 0.794 - 0.806 | ($p < 0.001$) |
| | Marital-status | **0.175** | 0.139 - 0.256 | $p < 0.001$ | **0.210** | 0.190 - 0.238 | $p < 0.001$ | 0.805 | 0.802 - 0.808 | ($p < 0.001$) |
| | Race | **0.115** | 0.049 - 0.219 | $p < 0.001$ | **0.054** | 0.033 - 0.076 | $p < 0.001$ | 0.807 | 0.803 - 0.810 | (0.005) |
| | Sex | **0.031** | 0.005 - 0.076 | $p < 0.001$ | **0.088** | 0.067 - 0.118 | $p < 0.001$ | 0.800 | 0.792 - 0.805 | ($p < 0.001$) |
| ARMED ME baseline | Age | 0.169 | 0.153 - 0.181 | | 0.146 | 0.139 - 0.157 | | | | |
| | Marital-status | 0.191 | 0.166 - 0.228 | | 0.212 | 0.201 - 0.228 | | 0.812 | 0.810 - 0.814 | |
| | Race | 0.140 | 0.089 - 0.187 | | 0.072 | 0.063 - 0.082 | | | | |
| | Sex | 0.098 | 0.087 - 0.110 | | 0.137 | 0.128 - 0.153 | | | | |
| Fair ARMED ME | Age | **0.077** | 0.013 - 0.138 | $p < 0.001$ | **0.103** | 0.063 - 0.143 | $p < 0.001$ | 0.803 | 0.797 - 0.810 | ($p < 0.001$) |
| | Marital-status | **0.147** | 0.109 - 0.235 | $p < 0.001$ | **0.180** | 0.152 - 0.214 | $p < 0.001$ | 0.807 | 0.803 - 0.810 | ($p < 0.001$) |
| | Race | **0.110** | 0.056 - 0.177 | $p < 0.001$ | **0.052** | 0.035 - 0.088 | $p < 0.001$ | 0.811 | 0.809 - 0.814 | $p < 0.001$ |
| | Sex | **0.017** | 0.001 - 0.057 | $p < 0.001$ | **0.070** | 0.047 - 0.110 | $p < 0.001$ | 0.801 | 0.796 - 0.811 | ($p < 0.001$) |
| | | | | Unseen occupations | | | | | | |
| ARMED FE baseline | Age | 0.220 | 0.181 - 0.238 | | 0.155 | 0.141 - 0.169 | | | | |
| | Marital-status | 0.243 | 0.215 - 0.271 | | 0.183 | 0.163 - 0.202 | | 0.773 | 0.760 - 0.778 | |
| | Race | 0.172 | 0.082 - 0.273 | | 0.056 | 0.038 - 0.077 | | | | |
| | Sex | 0.066 | 0.048 - 0.085 | | 0.098 | 0.085 - 0.112 | | | | |
| Fair ARMED FE | Age | **0.100** | 0.022 - 0.191 | $p < 0.001$ | **0.128** | 0.082 - 0.181 | $p < 0.001$ | 0.772 | 0.757 - 0.780 | 0.34 |
| | Marital-status | **0.232** | 0.199 - 0.280 | 0.02 | **0.175** | 0.151 - 0.198 | 0.01 | 0.771 | 0.763 - 0.780 | 0.06 |
| | Race | **0.151** | 0.073 - 0.281 | 0.11 | **0.054** | 0.035 - 0.085 | 0.50 | 0.771 | 0.762 - 0.779 | 0.11 |
| | Sex | **0.022** | 0.002 - 0.055 | $p < 0.001$ | **0.056** | 0.035 - 0.092 | $p < 0.001$ | 0.762 | 0.746 - 0.775 | ($p < 0.001$) |
| ARMED ME baselines | Age | 0.216 | 0.187 - 0.237 | | 0.151 | 0.136 - 0.163 | | | | |
| | Marital-status | 0.237 | 0.210 - 0.262 | | 0.175 | 0.156 - 0.194 | | 0.773 | 0.767 - 0.779 | |
| | Race | 0.164 | 0.069 - 0.245 | | **0.054** | 0.039 - 0.070 | | | | |
| | Sex | 0.064 | 0.050 - 0.084 | | 0.095 | 0.086 - 0.110 | | | | |
| Fair ARMED ME | Age | **0.087** | 0.015 - 0.175 | $p < 0.001$ | **0.111** | 0.055 - 0.160 | $p < 0.001$ | 0.768 | 0.754 - 0.779 | ($p < 0.001$) |
| | Marital-status | **0.219** | 0.187 - 0.276 | 0.001 | **0.152** | 0.125 - 0.189 | $p < 0.001$ | 0.765 | 0.752 - 0.780 | ($p < 0.001$) |
| | Race | **0.157** | 0.073 - 0.257 | 0.57 | 0.058 | 0.033 - 0.096 | 0.30 | 0.771 | 0.765 - 0.777 | (0.02) |
| | Sex | **0.036** | 0.009 - 0.061 | $p < 0.001$ | **0.043** | 0.025 - 0.083 | $p < 0.001$ | 0.756 | 0.743 - 0.774 | ($p < 0.001$) |

*p-values are measured against the corresponding baseline model*

*FE: Fixed Effects; ME: Mixed Effects; CI: conf. interval. The binary classification threshold was set via the Youden pt. from each training fold. The best fairness outcomes (min. std. dev. of classification accuracy across categories w/in each sensitive var.) are bolded. The conf. interval was established by training the mdl 40 times with different random seeds and a T-test to assess the mean of these samples.*

*Sex*, as shown in Table 4. For regression, equality odds fairness is measured through the standard deviations (SD) of the Mean Squared Error (MSE) and Root Mean Squared Error (RMSE) across the 5 age brackets for *Age* and across the 3 categories for *Sex*; a lower SD reflects greater fairness. For our framework's provider-invariant FE predictions (i.e., from the *fair ARMED FE* portion of the framework), *Age* shows an 11% improvement in MSE SD (reducing from 0.633 to 0.564) and an 11.4% improvement in RMSE SD. Meanwhile, the *Sex* variable in *fair ARMED FE* shows a 21% reduction in MSE SD and a 20% reduction in RMSE SD. All of these are statistically significant at $p < 0.001$.

For our framework's provider-specific ME predictions (i.e., from the *fair ARMED ME* portion of the framework), across *Age* brackets, there is a 7% reduction in MSE SD and a 6.8% reduction in RMSE SD. Meanwhile, across *Sex* categories, there is a 16% reduction in MSE SD and 15% reduction in RMSE SD, all significant at $p < 0.001$. **These reductions support the fairness-enhancing ARMED framework's ability to improve mixed effects fairness predictions across all fairness-sensitive variables for regression.**

## 4 DISCUSSION

Both the fair DA adversarial debiasing and the fair ARMED framework achieved significantly enhanced fairness. However, in the Adult dataset, the ARMED framework exhibits comparatively higher accuracy in both seen occupations (1.3% higher) and unseen occupations (1% higher), confirming the advantages of the full ARMED framework. Compared to non-fair models, a trade-off

Table 3: Prediction of the income level using the IPUMS dataset with two Debias head fairness enhancing ARMED framework.

| Model | Sensitive feature | TPR standard deviation | | | FPR standard deviation | | | Balanced accuracy (%) | | |
|---|---|---|---|---|---|---|---|---|---|---|
| | | Mean | 95% CI | p-value* | Mean | 95% CI | p-value* | Mean | 95% CI | p-value* |
| | | | | | Seen occupations | | | | | |
| ARMED FE baseline | Age | 0.161 | 0.153 - 0.171 | | 0.131 | 0.126 - 0.135 | | | | |
| | Marital-status | 0.072 | 0.070 - 0.076 | | 0.074 | 0.070 - 0.078 | | 0.788 | 0.787 - 0.789 | |
| | Race | 0.134 | 0.120 - 0.149 | | 0.107 | 0.098 - 0.129 | | | | |
| | Sex | 0.140 | 0.136 - 0.146 | | 0.147 | 0.143 - 0.151 | | | | |
| Fair ARMED FE | Age | **0.059** | 0.016 - 0.165 | $p < 0.001$ | **0.083** | 0.054 - 0.130 | $p < 0.001$ | 0.776 | 0.768 - 0.780 | ($p < 0.001$) |
| | Marital-status | **0.058** | 0.053 - 0.068 | $p < 0.001$ | **0.064** | 0.061 - 0.072 | $p < 0.001$ | 0.787 | 0.784 - 0.788 | ($p < 0.001$) |
| | Race | **0.077** | 0.055 - 0.125 | $p < 0.001$ | **0.055** | 0.043 - 0.088 | $p < 0.001$ | 0.786 | 0.785 - 0.788 | ($p < 0.001$) |
| | Sex | **0.090** | 0.081 - 0.121 | $p < 0.001$ | **0.101** | 0.090 - 0.133 | $p < 0.001$ | 0.784 | 0.782 - 0.787 | ($p < 0.001$) |
| ARMED ME baseline | Age | 0.164 | 0.155 - 0.173 | | 0.129 | 0.125 - 0.133 | | | | |
| | Marital-status | 0.070 | 0.067 - 0.074 | | 0.073 | 0.071 - 0.076 | | 0.794 | 0.793 - 0.795 | |
| | Race | 0.128 | 0.119 - 0.137 | | 0.105 | 0.097 - 0.123 | | | | |
| | Sex | 0.134 | 0.130 - 0.141 | | 0.146 | 0.141 - 0.150 | | | | |
| Fair ARMED ME | Age | **0.044** | 0.006 - 0.114 | $p < 0.001$ | **0.070** | 0.041 - 0.110 | $p < 0.001$ | 0.782 | 0.778 - 0.788 | ($p < 0.001$) |
| | Marital-status | **0.052** | 0.047 - 0.065 | $p < 0.001$ | **0.056** | 0.052 - 0.067 | $p < 0.001$ | 0.792 | 0.790 - 0.794 | $p < 0.001$ |
| | Race | **0.068** | 0.045 - 0.116 | $p < 0.001$ | **0.050** | 0.035 - 0.091 | $p < 0.001$ | 0.792 | 0.791 - 0.794 | $p < 0.001$ |
| | Sex | **0.065** | 0.050 - 0.113 | $p < 0.001$ | **0.078** | 0.061 - 0.124 | $p < 0.001$ | 0.787 | 0.785 - 0.793 | ($p < 0.001$) |
| | | | | | Unseen occupations | | | | | |
| ARMED FE baseline | Age | 0.261 | 0.249 - 0.276 | | 0.166 | 0.158 - 0.174 | | | | |
| | Marital-status | 0.127 | 0.118 - 0.135 | | 0.115 | 0.108 - 0.122 | | 0.724 | 0.720 - 0.727 | |
| | Race | 0.158 | 0.134 - 0.180 | | 0.095 | 0.080 - 0.110 | | | | |
| | Sex | 0.177 | 0.170 - 0.184 | | 0.133 | 0.127 - 0.140 | | | | |
| Fair ARMED FE | Age | **0.118** | 0.064 - 0.224 | $p < 0.001$ | **0.114** | 0.081 - 0.166 | $p < 0.001$ | 0.715 | 0.705 - 0.725 | ($p < 0.001$) |
| | Marital-status | **0.112** | 0.102 - 0.124 | $p < 0.001$ | **0.105** | 0.099 - 0.113 | $p < 0.001$ | 0.723 | 0.720 - 0.726 | ($p < 0.001$) |
| | Race | **0.100** | 0.072 - 0.151 | $p < 0.001$ | **0.063** | 0.053 - 0.084 | $p < 0.001$ | 0.725 | 0.722 - 0.729 | 0.07 |
| | Sex | **0.095** | 0.077 - 0.146 | $p < 0.001$ | **0.088** | 0.077 - 0.120 | $p < 0.001$ | 0.710 | 0.704 - 0.721 | ($p < 0.001$) |
| ARMED ME baseline | Age | 0.263 | 0.250 - 0.277 | | 0.168 | 0.162 - 0.175 | | | | |
| | Marital-status | 0.129 | 0.120 - 0.140 | | 0.117 | 0.113 - 0.124 | | 0.725 | 0.722 - 0.729 | |
| | Race | 0.158 | 0.138 - 0.181 | | 0.096 | 0.082 - 0.111 | | | | |
| | Sex | 0.181 | 0.169 - 0.188 | | 0.135 | 0.129 - 0.141 | | | | |
| Fair ARMED ME | Age | **0.100** | 0.043 - 0.197 | $p < 0.001$ | **0.103** | 0.071 - 0.165 | $p < 0.001$ | 0.713 | 0.704 - 0.725 | ($p < 0.001$) |
| | Marital-status | **0.104** | 0.092 - 0.122 | $p < 0.001$ | **0.100** | 0.094 - 0.110 | $p < 0.001$ | 0.722 | 0.718 - 0.727 | ($p < 0.001$) |
| | Race | **0.100** | 0.072 - 0.158 | $p < 0.001$ | **0.064** | 0.052 - 0.086 | $p < 0.001$ | 0.726 | 0.723 - 0.730 | 0.18 |
| | Sex | **0.063** | 0.033 - 0.140 | $p < 0.001$ | **0.070** | 0.054 - 0.112 | $p < 0.001$ | 0.702 | 0.694 - 0.720 | ($p < 0.001$) |

*p-values are measured against the corresponding baseline model*

*FE: Fixed Effects; ME: Mixed Effects; CI: conf. interval. The binary classification threshold was set via the Youden pt. from each training fold. The best fairness outcomes (min. std. dev. of classification accuracy across categories w/in each sensitive var.) are bolded. The conf. interval was established by training the model 40 times with different random seeds and a T-test to assess the mean of these samples.*

in accuracy versus fairness is expected, as fairness enhancement introduces an additional loss term to ensure the equality of odds. This means that the models aren't exclusively optimized for accuracy but strive for a balance between accuracy and fairness. Nevertheless, the average reduction in accuracy in our proposed full framework compared to non-fair models remains negligible, at less than 0.2%. Notably, the average trade-off is more pronounced in fair DA (1.1% in seen occupations and 1% in unseen occupations) compared to fair ARMED ME (0.15% in seen occupations and 0.175% in unseen occupations). There are variables with 95%CI overlaps between multiple methods, especially in the unseen clusters. This variability may stem from the current limitations of our z-predictor, especially when faced with complex site invariance. We are actively working on enhancing our z-predictor by refining hyperparameter optimization and increasing its complexity to better handle such scenarios. These improvements are slated for future development.

Yang et al. (2023)'s method shows consistent fairness improvement when predicting COVID data across various hospitals. However, the enhancement is relatively modest (below 5%) and the model is only demonstrated on classification tasks. In contrast, the fair ARMED framework consistently achieves much greater fairness enhancement across datasets from different sectors (finance, healthcare), and across both classification and regression tasks, with a substantial improvement in fairness of up to 83%. We suspect that this marked difference arises because we have taken more care in optimizing the trade-off between fairness and accuracy by exploring a greater number of hyperparameter configurations.

More importantly, our work preserves interpretability and advantages of the ARMED framework. For example with the ADULT dataset, the MEDL model uniquely distinguishes between occupation-

Table 4: Prediction (regression) of the number of hospitalization days using the Heritage Health dataset with the fairness enhancing ARMED framework

| Model | Sensitive feature | MSE standard deviation | | | RMSE standard deviation | | |
|---|---|---|---|---|---|---|---|
| | | Mean | 95% CI | p-value* | Mean | 95% CI | p-value* |
| | | | Seen providers | | | | |
| ARMED FE baseline | Age | 0.633 | 0.599 - 0.651 | | 0.289 | 0.275 - 0.296 | |
| | Sex | 0.323 | 0.292 - 0.342 | | 0.141 | 0.128 - 0.148 | |
| Fair ARMED FE | Age | **0.564** | 0.538 - 0.618 | $p < 0.001$ | **0.256** | 0.241 - 0.282 | $p < 0.001$ |
| | Sex | **0.256** | 0.240 - 0.327 | $p < 0.001$ | **0.113** | 0.107 - 0.142 | $p < 0.001$ |
| ARMED ME baseline | Age | 0.639 | 0.617 - 0.655 | | 0.292 | 0.282 - 0.297 | |
| | Sex | 0.329 | 0.309 - 0.342 | | 0.144 | 0.135 - 0.148 | |
| Fair ARMED ME | Age | **0.594** | 0.573 - 0.628 | $p < 0.001$ | **0.272** | 0.262 - 0.287 | $p < 0.001$ |
| | Sex | **0.277** | 0.265 - 0.333 | $p < 0.001$ | **0.123** | 0.117 - 0.145 | $p < 0.001$ |
| *p-values are measured against the corresponding baseline model | | | | | | | |

*FE: Fixed Effects; ME: Mixed Effects; CI: conf. interval. The best fairness outcomes (min. std. dev. of regression accuracy across categories w/in each sensitive var.) are bolded. The conf. interval was established by training the model 40 times with different random seeds and a T-test to assess the mean of these samples.*

independent effects, learned through the FE subnetwork, and occupation-specific effects, learned through the RE subnetwork. The integration of fairness into the MEDL framework preserves this distinctive capability while enhancing the fairness of each subnetwork. This dual approach enables our fair MEDL framework to provide insights into both the fair cluster-specific effects and the fair cluster-agnostic effects. This capability offers a deeper level of interpretability compared to conventional neural networks, which typically do not differentiate between cluster-specific and cluster-agnostic effects. Therefore, our proposed fair MEDL framework not only advances in terms of fairness but also in terms of interpretability, offering more nuanced insights into the model's learning process.

Our research demonstrates substantial improvements in fairness, with direct implications for everyday applications. The Adult and IPUMS datasets are illustrative of pivotal income predictions that machine learning models make, which play a role in critical financial decisions such as loan and credit approvals. By applying our framework, these processes can become significantly fairer, ensuring equitable treatment across diverse demographics. Similarly, the Heritage Health dataset focuses on predicting future hospital utilization for individuals, which plays a central role in the calculation of insurance premiums, typically a major expense. Fairness in these predictions is crucial, as it directly impacts individuals' financial obligations for healthcare. Thus, our framework not only enhances model fairness but also contributes to more equitable decision-making in crucial sectors affecting daily life.

## 5 CONCLUSION

We describe two key needs for improved machine learning including improved fairness and the handling of clustered data through explicit quantification of fixed, random, and mixed effects. We motivated the use of the ARMED framework, for mixed effects deep learning due to its improved performance and generalization to OOD data, improved interpretability, and ability to mitigate confounds, including Type 1 and 2 errors. We demonstrated how the combination of ARMED and domain adversarial debiasing method significantly boosts fairness for both the ARMED model and its ablation, the Domain Adversarial model. These improvements were consistent across three distinct datasets: two census datasets focused on income prediction and a healthcare dataset where the prediction target is the number of days an individual will spend hospitalized in the subsequent year. Such targets are commonly used in loan applications and Healthcare insurance decisions. Given the profound societal implications of these tasks, ensuring fairness is paramount. Our results strongly support the fairness enhancing ARMED adversarial debiasing approach, allowing stakeholders to reap the many salient benefits of MEDL while ensuring fairness for all individuals.

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

# A APPENDIX

## A.1 MODEL FIGURES

The architecture used for the models described in Section 2.2.2 are shown below. These include the domain adversarial (DA) neural network in Fig. 2, and the domain adversarial network with absolute correlation loss (fair DA ACL) in Fig. 3. The fair DA ACL model promotes fairness for one sensitive feature; however, this method can be extended to address several features simultaneously by computing multiple Pearson loss coefficients as encapsulated by the model's general loss function (Eq. 4).

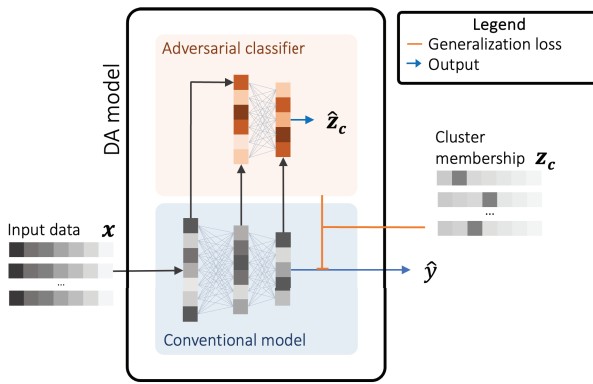

Figure 2: Domain Adversarial neural network, denoted as DA.

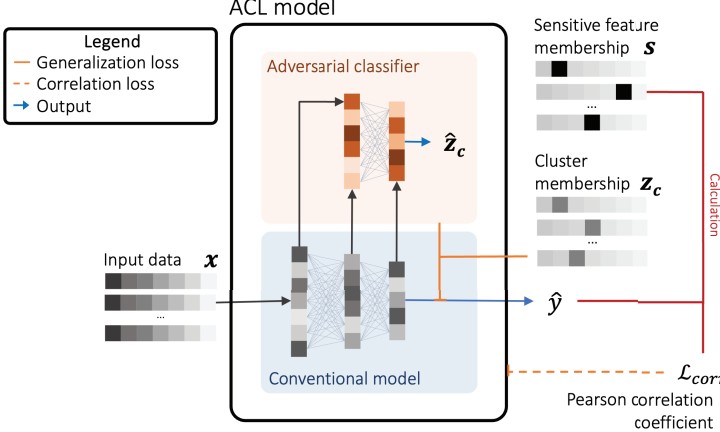

Figure 3: Domain Adversarial neural network with absolute correlation loss, denoted fair DA ACL.

## A.2 ADDITIONAL RESULTS

Table 5 presents the results of the domain adversarial variants tested on the IPUMS USA dataset. Both the fair DA ACL and fair DA adv. deb. models performed well, providing significant fairness increases. Further, the DA adv. deb. model here is a multi-headed implementation of adversarial debiasing, suggesting the feasibility of this approach and encouraging further exploration and refinement in future studies. The added hyperparameters and associated loss function complexity made this multi-headed model more challenging to tune. Additional tuning could likely yield further fairness increases beyond the results shown here.

The fairness-accuracy trade-off is an important consideration. Although the fair DA ACL model displays significant fairness improvements for *marital-status* and *sex* across both seen occupations

and unseen occupations, that particular model also has significantly lower balanced accuracy for samples with unseen occupations at 70.8%, a 2% drop from the baseline 72.8% balanced accuracy. In comparison, the fair DA adv. deb. model preserves much of the accuracy—even when extended to unseen occupations—while still providing statistically significant fairness increases for all sensitive features.

Table 5: Prediction of income level using the IPUMS dataset with the domain adversarial and comparable models

| Model | Sensitive feature | TPR standard deviation | | | FPR standard deviation | | | Balanced accuracy (%) | | |
|---|---|---|---|---|---|---|---|---|---|---|
| | | Mean | 95% CI | p-value* | Mean | 95% CI | p-value* | Mean | 95% CI | p-value* |
| | | | | | Seen occupations | | | | | |
| DFNN | Age | 0.155 | 0.152 - 0.158 | | 0.209 | 0.209 - 0.210 | | **0.789** | 0.789 - 0.790 | |
| | Marital-status | 0.071 | 0.068 - 0.073 | | 0.128 | 0.127 - 0.128 | | | | |
| | Race | 0.130 | 0.124 - 0.143 | | 0.205 | 0.203 - 0.208 | | | | |
| | Sex | 0.140 | 0.138 - 0.141 | | 0.309 | 0.308 - 0.309 | | | | |
| DA | Age | 0.151 | 0.144 - 0.158 | $p < 0.001$ | 0.209 | 0.203 - 0.210 | 0.36 | 0.789 | 0.781 - 0.790 | 0.55 |
| | Marital-status | 0.070 | 0.064 - 0.073 | 0.04 | 0.128 | 0.124 - 0.129 | 0.47 | | | |
| | Race | 0.127 | 0.116 - 0.138 | 0.04 | 0.206 | 0.198 - 0.210 | 0.44 | | | |
| | Sex | 0.135 | 0.130 - 0.137 | $p < 0.001$ | 0.306 | 0.298 - 0.308 | 0.02 | | | |
| fair DA ACL | Age | 0.150 | 0.144 - 0.155 | $p < 0.001$ | **0.206** | 0.205 - 0.207 | $p < 0.001$ | 0.783 | 0.783 - 0.784 | $p < 0.001$ |
| | Marital-status | **0.049** | 0.046 - 0.052 | $p < 0.001$ | **0.120** | 0.119 - 0.121 | $p < 0.001$ | | | |
| | Race | 0.120 | 0.113 - 0.131 | $p < 0.001$ | 0.208 | 0.205 - 0.210 | $p < 0.001$ | | | |
| | Sex | **0.098** | 0.096 - 0.102 | $p < 0.001$ | **0.289** | 0.287 - 0.290 | $p < 0.001$ | | | |
| fair DA adv. deb. | Age | **0.132** | 0.125 - 0.139 | $p < 0.001$ | 0.207 | 0.206 - 0.208 | $p < 0.001$ | 0.789 | 0.788 - 0.789 | $p < 0.001$ |
| | Marital-status | 0.059 | 0.056 - 0.063 | $p < 0.001$ | 0.126 | 0.125 - 0.126 | $p < 0.001$ | | | |
| | Race | **0.064** | 0.054 - 0.076 | $p < 0.001$ | **0.173** | 0.166 - 0.181 | $p < 0.001$ | | | |
| | Sex | 0.121 | 0.118 - 0.126 | $p < 0.001$ | 0.300 | 0.299 - 0.303 | $p < 0.001$ | | | |
| | | | | | Unseen occupations | | | | | |
| DFNN | Age | 0.253 | 0.248 - 0.259 | | 0.095 | 0.094 - 0.095 | | **0.728** | 0.727 - 0.729 | |
| | Marital-status | 0.127 | 0.124 - 0.129 | | 0.071 | 0.070 - 0.071 | | | | |
| | Race | 0.149 | 0.141 - 0.157 | | 0.081 | 0.079 - 0.084 | | | | |
| | Sex | 0.182 | 0.180 - 0.185 | | 0.077 | 0.077 - 0.078 | | | | |
| DA | Age | 0.250 | 0.238 - 0.256 | 0.005 | 0.094 | 0.091 - 0.094 | $p < 0.001$ | 0.727 | 0.720 - 0.729 | 0.22 |
| | Marital-status | 0.123 | 0.119 - 0.128 | $p < 0.001$ | 0.070 | 0.067 - 0.071 | $p < 0.001$ | | | |
| | Race | 0.149 | 0.135 - 0.161 | 0.79 | 0.082 | 0.078 - 0.085 | 0.22 | | | |
| | Sex | 0.172 | 0.166 - 0.176 | $p < 0.001$ | 0.076 | 0.073 - 0.077 | $p < 0.001$ | | | |
| fair DA ACL | Age | 0.233 | 0.227 - 0.239 | $p < 0.001$ | **0.081** | 0.080 - 0.083 | $p < 0.001$ | 0.708 | 0.706 - 0.711 | $p < 0.001$ |
| | Marital-status | **0.091** | 0.086 - 0.097 | $p < 0.001$ | **0.058** | 0.057 - 0.060 | $p < 0.001$ | | | |
| | Race | 0.155 | 0.145 - 0.167 | $p < 0.001$ | 0.083 | 0.080 - 0.086 | 0.001 | | | |
| | Sex | **0.089** | 0.082 - 0.096 | $p < 0.001$ | **0.061** | 0.060 - 0.063 | $p < 0.001$ | | | |
| fair DA adv. deb. | Age | **0.226** | 0.218 - 0.234 | $p < 0.001$ | 0.090 | 0.089 - 0.091 | $p < 0.001$ | 0.723 | 0.721 - 0.725 | $p < 0.001$ |
| | Marital-status | 0.119 | 0.116 - 0.124 | $p < 0.001$ | 0.067 | 0.066 - 0.068 | $p < 0.001$ | | | |
| | Race | **0.086** | 0.077 - 0.093 | $p < 0.001$ | **0.056** | 0.049 - 0.063 | $p < 0.001$ | | | |
| | Sex | 0.149 | 0.142 - 0.159 | $p < 0.001$ | 0.072 | 0.070 - 0.073 | $p < 0.001$ | | | |

*All p-values are measured against the baseline classifier (DFNN)*

*DFNN: dense feedforward neural network; DA: domain adversarial model; fair DA ACL: domain adv. mdl with absolute correlation loss for fairness enhancement; fair DA adv. deb.: domain adv mdl with adversarial debiasing for fairness enhancement; CI: conf. interval. Metrics were computed through 30 trials of each mdl across 10 training folds with different random seeds. Binary classification threshold was set via the Youden point determined at each training fold. The best fairness results (lowest std. dev. in classification accuracy across categories w/in each sensitive var.) are bolded.*

Table6 and 7 demonstrate results using DFNN and DFNN with debiasing head - Yang's methodYang et al. (2023), which aligns closely with our FixedEffect subnetwork's outcomes in table2 and 3, there is a mix in fairness enhancement but predictive accuracy are relatively simiar. This similarity bolsters our argument that the addition of the RandomEffect subnetwork marks a substantial improvement. By effectively managing batch effects, our model demonstrates enhanced capacity for fairness enhancement, as indicated in our revised results. Table8 and 9 demonstrate the impact of implementing Absolute Correlation Loss (ACL) in the ARMED framework as an alternative to debiasing heads. While ARMED ACL shows variability in fairness improvement across different sensitive features, it also leads to a significant decrease in prediction accuracy, reaching up to a 6% reduction. Considering these outcomes, we assert that debiasing heads, with its more uniform improvement in fairness and only minimal impact on prediction accuracy, is a preferable option compared to ACL at least in the 3 datasets we presented in this work.

Table 6: Prediction of income level using the ADULT dataset with the DFNN models

| Model | Sensitive feature | TPR standard deviation | | | FPR standard deviation | | | Balanced accuracy (%) | | |
|---|---|---|---|---|---|---|---|---|---|---|
| | | Mean | 95% CI | p-value* | Mean | 95% CI | p-value* | Mean | 95% CI | p-value* |
| | | | | | Seen occupations | | | | | |
| DFNN | Age | 0.170 | 0.154 - 0.188 | | 0.148 | 0.134 - 0.155 | | 0.809 | 0.806 - 0.811 | |
| | Marital-status | 0.201 | 0.173 - 0.226 | | 0.218 | 0.204 - 0.231 | | | | |
| | Race | 0.169 | 0.101 - 0.302 | | **0.077** | 0.065 - 0.097 | | | | |
| | Sex | 0.101 | 0.087 - 0.119 | | 0.139 | 0.127 - 0.150 | | | | |
| fair DFNN adv. deb. | Age | **0.059** | 0.009 - 0.172 | $p < 0.001$ | **0.091** | 0.047 - 0.151 | $p < 0.001$ | 0.795 | 0.784 - 0.809 | $(p < 0.001)$ |
| | Marital-status | **0.187** | 0.091 - 0.261 | 0.07 | **0.193** | 0.115 - 0.260 | $p < 0.001$ | 0.789 | 0.757 - 0.807 | $(p < 0.001)$ |
| | Race | **0.157** | 0.059 - 0.284 | 0.39 | 0.082 | 0.035 - 0.177 | 0.35 | 0.806 | 0.798 - 0.810 | $(p < 0.001)$ |
| | Sex | **0.012** | 0.001 - 0.053 | $p < 0.001$ | **0.062** | 0.047 - 0.102 | $p < 0.001$ | 0.796 | 0.790 - 0.803 | $(p < 0.001)$ |
| | | | | | Unseen sites | | | | | |
| DFNN | Age | 0.217 | 0.190 - 0.244 | | 0.150 | 0.129 - 0.165 | | 0.773 | 0.759 - 0.780 | |
| | Marital-status | 0.243 | 0.212 - 0.270 | | 0.176 | 0.157 - 0.190 | | | | |
| | Race | **0.182** | 0.077 - 0.286 | | **0.059** | 0.044 - 0.082 | | | | |
| | Sex | 0.064 | 0.037 - 0.088 | | 0.095 | 0.079 - 0.105 | | | | |
| fair DFNN adv. deb. | Age | **0.069** | 0.012 - 0.223 | $p < 0.001$ | **0.087** | 0.032 - 0.156 | $p < 0.001$ | 0.761 | 0.726 - 0.778 | $(p < 0.001)$ |
| | Marital-status | **0.237** | 0.164 - 0.304 | 0.45 | **0.161** | 0.070 - 0.240 | 0.08 | 0.743 | 0.676 - 0.775 | $(p < 0.001)$ |
| | Race | 0.202 | 0.071 - 0.325 | 0.24 | 0.073 | 0.032 - 0.144 | (0.004) | 0.771 | 0.757 - 0.781 | 0.23 |
| | Sex | **0.051** | 0.014 - 0.081 | $p < 0.001$ | **0.034** | 0.016 - 0.069 | $p < 0.001$ | 0.748 | 0.726 - 0.768 | $(p < 0.001)$ |

*p-values are measured against the corresponding baseline model*

*DFNN: Deep Feedforward Neural Network; CI: conf. interval. The binary classification threshold was set via the Youden pt. from each training fold. The best fairness outcomes (min. std. dev. of classification accuracy across categories w/in each sensitive var.) are bolded. The conf. interval was established by training the mdl 40 times with different random seeds and a T-test to assess the mean of these samples.*

Table 7: Prediction of income level using the IPUMS dataset with the DFNN models

| Model | Sensitive feature | TPR standard deviation | | | FPR standard deviation | | | Balanced accuracy (%) | | |
|---|---|---|---|---|---|---|---|---|---|---|
| | | Mean | 95% CI | p-value* | Mean | 95% CI | p-value* | Mean | 95% CI | p-value* |
| | | | | | Seen occupations | | | | | |
| DFNN | Age | 0.161 | 0.147 - 0.168 | | 0.131 | 0.121 - 0.135 | | 0.787 | 0.787 - 0.788 | |
| | Marital-status | 0.072 | 0.068 - 0.076 | | 0.074 | 0.068 - 0.077 | | | | |
| | Race | 0.136 | 0.125 - 0.148 | | 0.110 | 0.097 - 0.136 | | | | |
| | Sex | 0.140 | 0.132 - 0.147 | | 0.147 | 0.142 - 0.151 | | | | |
| fair DFNN adv. deb. | Age | **0.027** | 0.004 - 0.161 | $p < 0.001$ | **0.060** | 0.043 - 0.131 | $p < 0.001$ | 0.775 | 0.771 - 0.788 | $(p < 0.001)$ |
| | Marital-status | **0.060** | 0.042 - 0.075 | $p < 0.001$ | **0.050** | 0.035 - 0.082 | $p < 0.001$ | 0.783 | 0.781 - 0.787 | $(p < 0.001)$ |
| | Race | **0.103** | 0.055 - 0.203 | $p < 0.001$ | **0.070** | 0.037 - 0.106 | $p < 0.001$ | 0.786 | 0.782 - 0.788 | $(p < 0.001)$ |
| | Sex | **0.050** | 0.034 - 0.135 | $p < 0.001$ | **0.058** | 0.042 - 0.139 | $p < 0.001$ | 0.777 | 0.774 - 0.787 | $(p < 0.001)$ |
| | | | | | Unseen sites | | | | | |
| DFNN | Age | 0.256 | 0.242 - 0.270 | | 0.165 | 0.153 - 0.173 | | 0.724 | 0.721 - 0.726 | |
| | Marital-status | 0.127 | 0.119 - 0.136 | | 0.115 | 0.105 - 0.121 | | | | |
| | Race | 0.160 | 0.141 - 0.185 | | 0.096 | 0.080 - 0.114 | | | | |
| | Sex | 0.176 | 0.168 - 0.183 | | 0.133 | 0.125 - 0.140 | | | | |
| fair DFNN adv. deb. | Age | **0.082** | 0.044 - 0.259 | $p < 0.001$ | **0.088** | 0.069 - 0.166 | $p < 0.001$ | 0.706 | 0.697 - 0.725 | $(p < 0.001)$ |
| | Marital-status | **0.093** | 0.072 - 0.129 | $p < 0.001$ | **0.088** | 0.073 - 0.115 | $p < 0.001$ | 0.716 | 0.709 - 0.722 | $(p < 0.001)$ |
| | Race | **0.131** | 0.077 - 0.211 | $p < 0.001$ | **0.078** | 0.054 - 0.112 | $p < 0.001$ | 0.724 | 0.720 - 0.726 | 0.62 |
| | Sex | **0.023** | 0.000 - 0.164 | $p < 0.001$ | **0.048** | 0.032 - 0.128 | $p < 0.001$ | 0.687 | 0.677 - 0.723 | $(p < 0.001)$ |

*p-values are measured against the corresponding baseline model*

*DFNN: Deep Feedforward Neural Network; CI: conf. interval. The binary classification threshold was set via the Youden pt. from each training fold. The best fairness outcomes (min. std. dev. of classification accuracy across categories w/in each sensitive var.) are bolded. The conf. interval was established by training the mdl 40 times with different random seeds and a T-test to assess the mean of these samples.*

Figures 4, 5 and 6 illustrate the balance between accuracy and enhanced fairness in our model. We observed substantial improvements in both TPR SD and FPR SD, with fairness enhancements reaching up to 67% at a minimal accuracy trade-off of about 1%. Notably, the MixedEffect (ME) models outperform their FixedEffect (FE) counterparts in both accuracy and fairness, underscoring the efficacy of MixedEffect models in balancing fairness with accuracy. We recognize that the larger standard deviations in fairness and accuracy metrics for ARMED models could be attributed to their complexity, resulting in broader convergence areas post-training.

Table 8: Prediction of income level using the ADULT dataset with the fairness enhancing ARMED ACL.

| Model | Sensitive feature | TPR standard deviation | | | FPR standard deviation | | | Balanced accuracy (%) | | |
|---|---|---|---|---|---|---|---|---|---|---|
| | | Mean | 95% CI | p-value* | Mean | 95% CI | p-value* | Mean | 95% CI | p-value* |
| | | | | | Seen occupations | | | | | |
| ARMED FE baseline | Age | 0.175 | 0.161 - 0.188 | | 0.149 | 0.140 - 0.161 | | | | |
| | Marital-status | **0.210** | 0.186 - 0.254 | | **0.224** | 0.203 - 0.237 | | 0.808 | 0.806 - 0.810 | |
| | Race | 0.158 | 0.099 - 0.229 | | **0.076** | 0.066 - 0.093 | | | | |
| | Sex | 0.106 | 0.091 - 0.121 | | 0.141 | 0.127 - 0.156 | | | | |
| ARMED ACL FE | Age | **0.043** | 0.028 - 0.070 | $p < 0.001$ | **0.081** | 0.056 - 0.091 | $p < 0.001$ | 0.790 | 0.779 - 0.796 | $(p < 0.001)$ |
| | Marital-status | 0.295 | 0.207 - 0.350 | $(p < 0.001)$ | 0.268 | 0.111 - 0.339 | $(p < 0.001)$ | 0.761 | 0.693 - 0.781 | $(p < 0.001)$ |
| | Race | **0.127** | 0.069 - 0.202 | $p < 0.001$ | 0.117 | 0.084 - 0.202 | $(p < 0.001)$ | 0.806 | 0.804 - 0.808 | $(p < 0.001)$ |
| | Sex | **0.037** | 0.017 - 0.064 | $p < 0.001$ | **0.017** | 0.002 - 0.033 | $p < 0.001$ | 0.775 | 0.765 - 0.785 | $(p < 0.001)$ |
| ARMED ME baseline | Age | 0.170 | 0.155 - 0.185 | | 0.144 | 0.135 - 0.154 | | | | |
| | Marital-status | **0.194** | 0.172 - 0.233 | | **0.210** | 0.191 - 0.225 | | 0.812 | 0.810 - 0.814 | |
| | Race | 0.143 | 0.094 - 0.188 | | **0.072** | 0.062 - 0.082 | | | | |
| | Sex | 0.099 | 0.089 - 0.112 | | 0.136 | 0.124 - 0.151 | | | | |
| ARMED ACL ME | Age | **0.062** | 0.050 - 0.074 | $p < 0.001$ | **0.062** | 0.048 - 0.073 | $p < 0.001$ | 0.780 | 0.773 - 0.785 | $(p < 0.001)$ |
| | Marital-status | 0.277 | 0.202 - 0.338 | $(p < 0.001)$ | 0.247 | 0.133 - 0.315 | $(p < 0.001)$ | 0.766 | 0.739 - 0.778 | $(p < 0.001)$ |
| | Race | **0.113** | 0.072 - 0.165 | $p < 0.001$ | 0.124 | 0.091 - 0.169 | $(p < 0.001)$ | 0.810 | 0.808 - 0.812 | $p < 0.001$ |
| | Sex | **0.046** | 0.028 - 0.062 | $p < 0.001$ | **0.007** | 0.000 - 0.019 | $p < 0.001$ | 0.777 | 0.770 - 0.785 | $(p < 0.001)$ |
| | | | | | Unseen occupations | | | | | |
| ARMED FE baseline | Age | 0.222 | 0.196 - 0.245 | | 0.154 | 0.136 - 0.168 | | | | |
| | Marital-status | **0.246** | 0.217 - 0.267 | | **0.182** | 0.150 - 0.198 | | 0.771 | 0.761 - 0.778 | |
| | Race | 0.179 | 0.086 - 0.244 | | **0.056** | 0.038 - 0.079 | | | | |
| | Sex | **0.065** | 0.040 - 0.085 | | 0.097 | 0.077 - 0.109 | | | | |
| ARMED ACL FE | Age | **0.060** | 0.023 - 0.104 | $p < 0.001$ | **0.088** | 0.068 - 0.106 | $p < 0.001$ | 0.750 | 0.726 - 0.766 | $(p < 0.001)$ |
| | Marital-status | 0.337 | 0.166 - 0.419 | $(p < 0.001)$ | 0.233 | 0.072 - 0.323 | $(p < 0.001)$ | 0.711 | 0.663 - 0.739 | $(p < 0.001)$ |
| | Race | **0.104** | 0.032 - 0.231 | $p < 0.001$ | 0.097 | 0.051 - 0.213 | $(p < 0.001)$ | 0.770 | 0.760 - 0.777 | 0.17 |
| | Sex | 0.103 | 0.064 - 0.158 | $(p < 0.001)$ | **0.007** | 0.001 - 0.021 | $p < 0.001$ | 0.719 | 0.676 - 0.743 | $(p < 0.001)$ |
| ARMED ME baseline | Age | 0.219 | 0.194 - 0.238 | | 0.149 | 0.133 - 0.165 | | | | |
| | Marital-status | **0.239** | 0.210 - 0.263 | | **0.173** | 0.147 - 0.188 | | 0.772 | 0.760 - 0.780 | |
| | Race | 0.169 | 0.082 - 0.233 | | **0.054** | 0.038 - 0.072 | | | | |
| | Sex | **0.064** | 0.044 - 0.081 | | 0.094 | 0.082 - 0.107 | | | | |
| ARMED ACL ME | Age | **0.140** | 0.106 - 0.170 | $p < 0.001$ | **0.083** | 0.054 - 0.102 | $p < 0.001$ | 0.716 | 0.699 - 0.728 | $(p < 0.001)$ |
| | Marital-status | 0.335 | 0.198 - 0.418 | $(p < 0.001)$ | 0.224 | 0.089 - 0.328 | $(p < 0.001)$ | 0.714 | 0.679 - 0.737 | $(p < 0.001)$ |
| | Race | **0.092** | 0.033 - 0.227 | $p < 0.001$ | 0.105 | 0.064 - 0.200 | $(p < 0.001)$ | 0.769 | 0.761 - 0.777 | $(0.01)$ |
| | Sex | 0.118 | 0.088 - 0.161 | $(p < 0.001)$ | **0.008** | 0.000 - 0.025 | $p < 0.001$ | 0.715 | 0.685 - 0.739 | $(p < 0.001)$ |

*p-values are measured against the corresponding baseline model

*FE: Fixed Effects; ME: Mixed Effects; ACL: Absolute Correlation loss; CI: conf. interval. The binary classification threshold was set via the Youden pt. from each training fold. The best fairness outcomes (min. std. dev. of classification accuracy across categories w/in each sensitive var.) are bolded. The conf. interval was established by training the mdl 40 times with different random seeds and a T-test to assess the mean of these samples.*

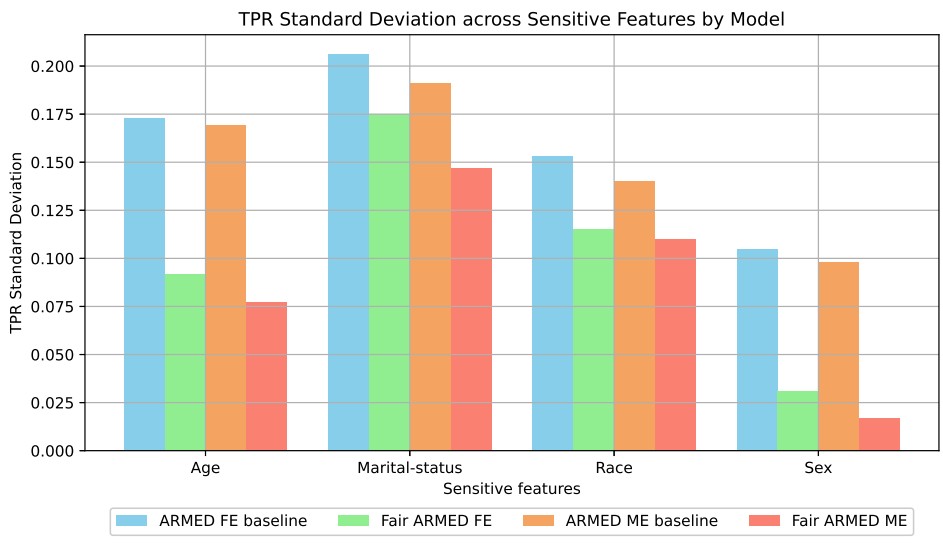

Figure 4: IPUMS dataset, True Positive Rate (TPR) standard deviation of different models and sensitive features on seen occupations.

Table 9: Prediction of income level using the IPUMS dataset with the fairness enhancing ARMED ACL.

| Model | Sensitive feature | TPR standard deviation | | | FPR standard deviation | | | Balanced accuracy (%) | | |
|---|---|---|---|---|---|---|---|---|---|---|
| | | Mean | 95% CI | p-value* | Mean | 95% CI | p-value* | Mean | 95% CI | p-value* |
| | | | | | Seen occupations | | | | | |
| ARMED FE baseline | Age | **0.161** | 0.153 - 0.170 | | **0.131** | 0.128 - 0.135 | | 0.788 | 0.787 - 0.789 | |
| | Marital-status | **0.072** | 0.069 - 0.076 | | **0.074** | 0.071 - 0.077 | | | | |
| | Race | 0.134 | 0.122 - 0.150 | | **0.108** | 0.097 - 0.128 | | | | |
| | Sex | 0.140 | 0.135 - 0.145 | | 0.148 | 0.144 - 0.151 | | | | |
| ARMED ACL FE | Age | 0.179 | 0.168 - 0.191 | $(p < 0.001)$ | 0.137 | 0.134 - 0.142 | $(p < 0.001)$ | 0.782 | 0.781 - 0.783 | $(p < 0.001)$ |
| | Marital-status | 0.085 | 0.073 - 0.105 | $(p < 0.001)$ | 0.099 | 0.074 - 0.130 | $(p < 0.001)$ | 0.763 | 0.757 - 0.768 | $(p < 0.001)$ |
| | Race | **0.120** | 0.107 - 0.129 | $p < 0.001$ | 0.123 | 0.110 - 0.141 | $(p < 0.001)$ | 0.785 | 0.784 - 0.786 | $(p < 0.001)$ |
| | Sex | **0.050** | 0.034 - 0.065 | $p < 0.001$ | **0.056** | 0.037 - 0.073 | $p < 0.001$ | 0.772 | 0.763 - 0.778 | $(p < 0.001)$ |
| ARMED ME baseline | Age | **0.164** | 0.154 - 0.174 | | **0.129** | 0.126 - 0.133 | | 0.794 | 0.793 - 0.794 | |
| | Marital-status | **0.070** | 0.067 - 0.074 | | **0.073** | 0.071 - 0.077 | | | | |
| | Race | 0.128 | 0.118 - 0.135 | | **0.105** | 0.097 - 0.121 | | | | |
| | Sex | 0.134 | 0.129 - 0.139 | | 0.146 | 0.141 - 0.150 | | | | |
| ARMED ACL ME | Age | **0.179** | 0.167 - 0.189 | $(p < 0.001)$ | **0.136** | 0.132 - 0.141 | $(p < 0.001)$ | 0.789 | 0.788 - 0.790 | $p < 0.001$ |
| | Marital-status | 0.076 | 0.062 - 0.090 | $(p < 0.001)$ | 0.099 | 0.075 - 0.127 | $(p < 0.001)$ | 0.770 | 0.766 - 0.773 | $(p < 0.001)$ |
| | Race | **0.113** | 0.102 - 0.123 | $p < 0.001$ | 0.117 | 0.108 - 0.132 | $(p < 0.001)$ | 0.791 | 0.790 - 0.792 | $p < 0.001$ |
| | Sex | **0.023** | 0.005 - 0.039 | $p < 0.001$ | **0.014** | 0.001 - 0.028 | $p < 0.001$ | 0.756 | 0.752 - 0.762 | $(p < 0.001)$ |
| | | | | | Unseen occupations | | | | | |
| ARMED FE baseline | Age | 0.262 | 0.249 - 0.277 | | 0.167 | 0.158 - 0.175 | | 0.724 | 0.721 - 0.728 | |
| | Marital-status | **0.126** | 0.117 - 0.134 | | **0.115** | 0.110 - 0.121 | | | | |
| | Race | 0.158 | 0.132 - 0.181 | | **0.095** | 0.076 - 0.111 | | | | |
| | Sex | 0.177 | 0.169 - 0.185 | | 0.133 | 0.126 - 0.140 | | | | |
| ARMED ACL FE | Age | **0.256** | 0.236 - 0.279 | 0.02 | **0.158** | 0.147 - 0.168 | $p < 0.001$ | 0.721 | 0.716 - 0.726 | $(p < 0.001)$ |
| | Marital-status | 0.137 | 0.111 - 0.162 | $(p < 0.001)$ | 0.116 | 0.090 - 0.143 | 0.77 | 0.685 | 0.673 - 0.694 | $(p < 0.001)$ |
| | Race | **0.145** | 0.123 - 0.168 | $p < 0.001$ | 0.110 | 0.094 - 0.142 | $(p < 0.001)$ | 0.721 | 0.717 - 0.725 | $(p < 0.001)$ |
| | Sex | **0.022** | 0.001 - 0.050 | $p < 0.001$ | 0.043 | 0.021 - 0.063 | $p < 0.001$ | 0.685 | 0.665 - 0.700 | $(p < 0.001)$ |
| ARMED ME baseline | Age | 0.264 | 0.250 - 0.280 | | 0.168 | 0.160 - 0.174 | | 0.725 | 0.722 - 0.729 | |
| | Marital-status | **0.128** | 0.118 - 0.138 | | **0.117** | 0.112 - 0.122 | | | | |
| | Race | 0.160 | 0.139 - 0.184 | | **0.097** | 0.080 - 0.115 | | | | |
| | Sex | 0.181 | 0.173 - 0.190 | | 0.135 | 0.128 - 0.143 | | | | |
| ARMED ACL ME | Age | **0.260** | 0.235 - 0.282 | 0.08 | **0.162** | 0.153 - 0.170 | $p < 0.001$ | 0.724 | 0.719 - 0.728 | 0.007 |
| | Marital-status | 0.134 | 0.108 - 0.155 | (0.02) | 0.119 | 0.094 - 0.142 | 0.35 | 0.690 | 0.681 - 0.697 | $(p < 0.001)$ |
| | Race | **0.144** | 0.124 - 0.170 | $p < 0.001$ | 0.107 | 0.091 - 0.141 | $(p < 0.001)$ | 0.723 | 0.719 - 0.726 | $(p < 0.001)$ |
| | Sex | **0.093** | 0.070 - 0.120 | $p < 0.001$ | **0.011** | 0.002 - 0.021 | $p < 0.001$ | 0.649 | 0.638 - 0.662 | $(p < 0.001)$ |

*p-values are measured against the corresponding baseline model*

*FE: Fixed Effects; ME: Mixed Effects; ACL: Absolute Correlation loss; CI: conf. interval. The binary classification threshold was set via the Youden pt. from each training fold. The best fairness outcomes (min. std. dev. of classification accuracy across categories w/in each sensitive var.) are bolded. The conf. interval was established by training the mdl 40 times with different random seeds and a T-test to assess the mean of these samples.*

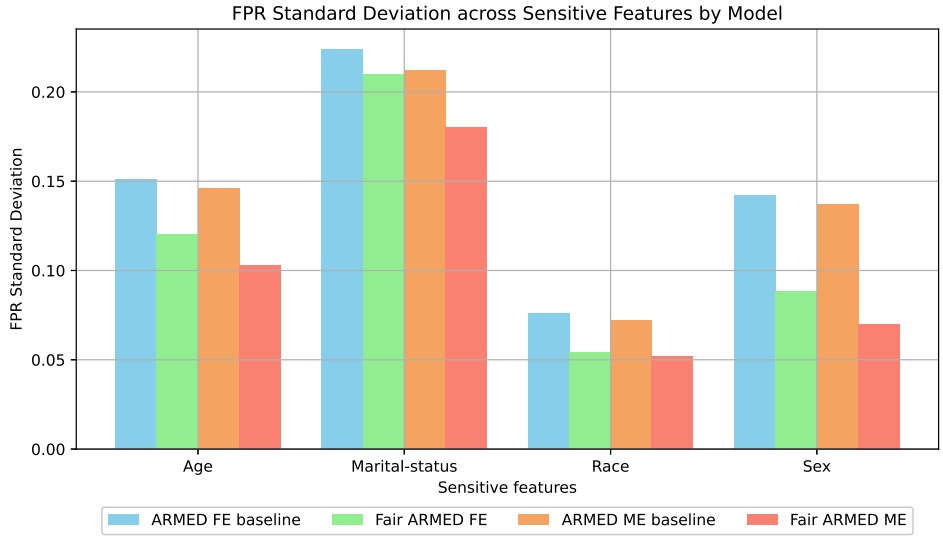

Figure 5: IPUMS dataset, False Positive Rate (FPR) standard deviation of different models and sensitive features on seen occupations.

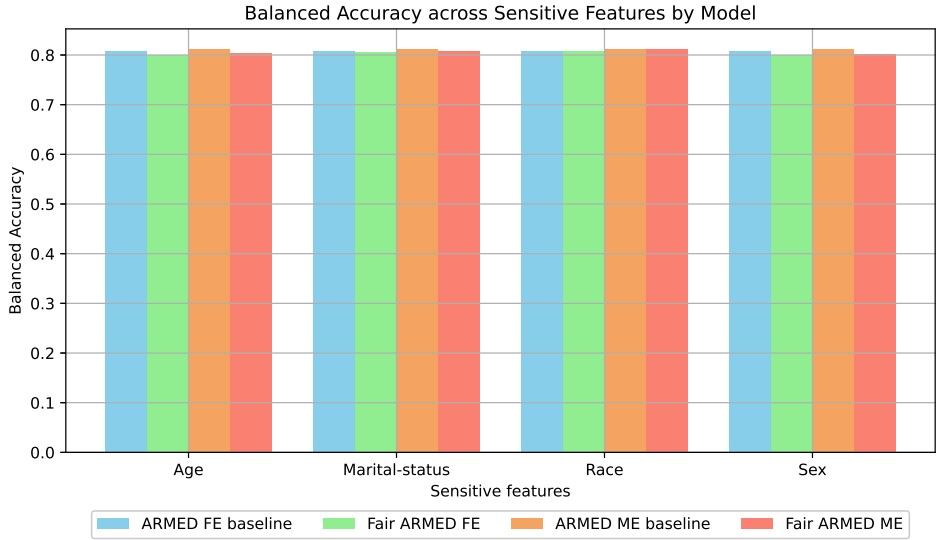

Figure 6: IPUMS dataset, balanced accuracy of different models and sensitive features on seen occupations.

