# OpenReview forum: "Fairness-enhancing mixed effects deep learning improves fairness on in- and out-of-distribution clustered (non-iid) data"
_ICLR.cc/2024/Conference — Submitted to ICLR 2024_

### Official Review · Reviewer_ejZU · 2023-10-23

**Soundness:** 2 fair
**Presentation:** 2 fair
**Contribution:** 2 fair
**Rating:** 3
**Confidence:** 4

**Summary:**

This paper extends the previous ARMED framework with adversarial neural networks to enhance fairness. It shows improvements in fairness across sensitive variables in various datasets.

**Strengths:**

1. The paper has tested the proposed method on a diverse set of real-world datasets from finance and medicine, showing improvements in fairness.

**Weaknesses:**

1. The paper is very poorly written. The structure is not well-presented, and there are many grammar errors.
2. The fairness issue is addressed with domain adversarial neural networks, which is common.
3. The mathematical definitions are unclear. Only several losses are introduced without any detailed interpretation.
4. It's unclear how the proposed method identifies new unseen clusters with an adversarial classifier.
5. There is no justification for why the proposed method can improve equality-of-odds fairness.

**Questions:**

Please see the previous section.

**Details Of Ethics Concerns:**

As mentioned by Reviewer vJ3t, this paper has violated the anonymity.

---

> ### Author Response · Authors · 2023-11-20
> **Response to Reviewer ejZU**
>
> Thank you for your detailed review and valuable feedback. We have taken your comments into consideration and made several revisions to address the issues raised:
>
> 1. We have made substantial efforts to enhance the clarity of our paper. All of our figures have been refined, new figures 4, 5, and 6 were added. Tables 2 and 3 were updated with two debiasing heads models and new tables 6,7,8,9 were added. Many sections were rewritten for better understanding. We welcome any remaining specific suggestions to improve the paper including grammar modifications.
>
> 2. While the use of debiasing heads (an adversarial approach) to promote fairness is indeed common in neural networks, their application and suitability to a MixedEffect deep learning framework is novel, and how to do this is non-obvious. Should the FE subnetwork be made fair? Should the RE subnetwork be made fair? Should both be made fair? How to trade off these constraints?  We address these core issues in this paper. This allows us to handle both confounding from cluster effects and improve fairness. Furthermore, we also test Absolute Correlation Loss (ACL) which is distinct from the debiasing heads methods, showcasing our study’s novelty.
>
> 3. We have reviewed our mathematical formulations and believe they are correctly presented. However, we are open to clarifying any specific equations or definitions you find ambiguous, and we will adjust our manuscript accordingly.
>
> 4. To clarify, our model uses an adversarial approach to promote fairness, specifically promoting equalized odds fairness. The cluster adversarial network aims to eliminate cluster-specific biases. The z-predictor is a non-adversarial, separately trained predictive model. It maps samples from clusters unseen during training, to a weighted combination of the most similar clusters seen during training. This allows the seen clusters to regularize the interpretation of the sample from the unseen cluster. It allows the MEDL framework to leverage the cluster specific models learned by the RandomEffects subnetwork. This mechanism is detailed in Figure 1.
>
> 5. Our choice of metrics for enforcing equalized odds fairness namely the reduction in standard deviations of TPR and FPR across sites, is grounded in previous literature [1,2]. Specifically, this approach was developed to alleviate the limitations of other measures of fairness. The approach also extends readily to regression (not just classification) and it can be implemented as a neural network. These metrics have been discussed in Section 2.1, and our empirical results demonstrate a significant reduction in these standard deviations, affirming the effectiveness of our approach.
>
> Regarding ethics concerns, we have thoroughly revised our figures to make it abundantly clear that our figures are original figures drawn by us from scratch.
>
> We sincerely hope that our revisions and clarifications address your concerns effectively.
>
> References
>
> [1] Moritz Hardt, Eric Price, and Nathan Srebro. Equality of opportunity in supervised learning. In 30th Conference on Neural Information Processing Systems (NIPS 2016), 2016.
>
> [2] Brian Hu Zhang, Blake Lemoine, and Margaret Mitchell. Mitigating unwanted biases with adversarial learning. In Proceedings of the 2018 AAAI/ACM Conference on AI, Ethics, and Society, pp. 335–340, 2018.

---

### Official Review · Reviewer_GZ1m · 2023-10-31

**Soundness:** 2 fair
**Presentation:** 2 fair
**Contribution:** 1 poor
**Rating:** 3
**Confidence:** 4

**Summary:**

The research paper presents an enhancement to the ARMED framework, aiming to improve fairness concerning fairness-sensitive variables like age, sex, and race. Although the original ARMED framework provided commendable generalization for out-of-distribution (OOD) data, it suffered from biases towards predominant groups. To address this, the authors incorporated adversarial debiasing (adv. deb.) and absolute correlation loss (ACL) into the existing domain adversarial (DA) model, a component of the ARMED framework.

Three distinct models were compared: the DA model, the fair DA adv. deb. model, and the DA ACL model. Among these, the fair DA adv. deb. model consistently exhibited enhanced fairness. Consequently, by integrating these modifications for both fixed and mixed effects, the authors devised a Fairness-enhancing ARMED framework. This refined model not only maintained a similar accuracy to the existing ARMED baseline but also exhibited considerable fairness improvements for classification tasks and better mixed effects fairness predictions in regression.

The study further showcased the model's efficacy using three datasets: the ADULT dataset, IPUMS dataset, and Heritage Health dataset. When pitted against existing neural networks, the "Fair" model, which synergized components from both the DA and ARMED models, produced superior fairness predictions across nearly every fairness-sensitive variable.

A significant contribution highlighted in the paper is the inclusion of sub-networks, specifically the Adversarial Classifier A_F and the Adversarial Classifier A_m, to the ARMED framework. The paper's findings serve as foundational work, promoting enhanced fairness and reliability in machine learning outputs, particularly in handling OOD data and emphasizing mixed effects fairness predictions in regression scenarios.

**Strengths:**

Originality: The paper addresses the pressing issue of bias in deep learning models, especially when it comes to fairness-sensitive variables. By melding the ARMED framework with domain adversarial techniques, the research manages to elevate both fairness and reliability in its predictions.

Quality: A prominent improvement is evident through a significant reduction in standard deviation , reflecting the enhanced quality of the model. The proposed model, which synergizes ARMED and DA techniques, exhibits superior performance when dealing with out-of-distribution data and fairness-sensitive variables.

Clarity: The research offers insights into the application and outcomes of various methods.

Significance: Manage the challenge of out-of-distribution data in deep learning, while ensuring fairness, accentuates its pivotal role in the advancement of the field.

**Weaknesses:**

Originality:
- The paper seems to predominantly enhance the existing ARMED model by integrating two adversarial debasing components. The modification, while valuable, might not be perceived as groundbreaking, especially when viewed against the backdrop of the existing literature.
- The work appears to be an iteration of the ARMED model rather than a transformative leap, raising concerns about the overall novelty and the magnitude of the paper's impact.

Clarity:
- Several figures and illustrations used in the paper closely resemble those from the original ARMED paper. This reuse of content, without adequate new context, can create confusion.
- Some terms, which although might be secondary in the context of this paper, are left undefined. Terms like "h", "𝛽", and "m", for instance, need clear explanations or references, even if briefly, to maintain reader continuity.
- The introduction contains superfluous discussions on traditional deep learning's weaknesses and excessive literature references with insufficient explanations. This can dilute the paper's main message and confuse readers.
- Explanations on the merging of the ARMED and Domain Adversarial models, particularly in Figure 1, are vague. A detailed breakdown or a supplementary diagram could enhance clarity.

Quality:
- The paper seems to underemphasize the importance of well-established fairness criteria. Metrics like equalized odds or demographic parity should be discussed more prominently, rather than lesser-known metrics such as TPR stdev or FPR stdev's mean and CI.
- Although the results are analyzed thoroughly, the overall experimental setup and methodology appear to lack depth. Without a comprehensive understanding of the experiment's design and employed metrics, the derived results might seem less credible.
- There's an observable absence of detailed visualizations in the experimental sections, reducing the impact and clarity of results presented.
- Explanations for certain core concepts like Fixed effect, Random effect, and Mixed effect are either missing or insufficiently highlighted. Such crucial components warrant a dedicated section, possibly in the introduction or an appendix.

Significance:
- While performance improvements are highlighted, the paper could benefit from a more persuasive argument showcasing how significant these improvements are in the larger context.
- Drawing direct visual and textual comparisons to the older ARMED model, without differentiating the advancements made in the current paper, might diminish its perceived value.

**Questions:**

- How does the addition of two adversarial debasing components differentiate your work from the original ARMED model substantially? Could you elaborate on the unique challenges and solutions introduced in this iteration?
- In the figures that resemble those from the original ARMED paper, are there any significant alterations or modifications that readers should be aware of? If so, could these be highlighted or differentiated more clearly?
- What led to the decision to focus on metrics like TPR stdev or FPR stdev's mean and CI instead of the more conventionally used fairness metrics such as equalized odds or demographic parity? How do the chosen metrics enhance the study's objectives?
- Is it possible to incorporate more detailed visualizations in the experimental sections to enhance clarity and understanding of the results? (I've seen curves which includes two axes - performance and fairness)
- Given the stated performance improvements, could you contextualize them more persuasively? How do these improvements translate to real-world applications or the larger academic context?
- There seems to be a strong emphasis on the paper's results. Could you provide more comprehensive background on the relevance and significance of these results in the context of existing research or practical applications?

**Details Of Ethics Concerns:**

Yes, Research integrity issues (e.g., plagiarism, dual submission)

Reason: Using the same figure from related work, especially without proper attribution or permission, can raise concerns about research integrity. Even if the related work is from the same lab or the same set of authors, it's essential to clearly cite the source and ensure there's no ambiguity about the origin of the figure. Repurposing figures or content without proper acknowledgment can be viewed as a form of self-plagiarism. It's vital for authors to ensure they have the right to use such figures and provide proper citations to maintain transparency and integrity in their research.

The primary illustration (Figure 1) of this manuscript appears to be directly sourced from a prior publication by the same group [1]. It would be helpful to contrast this with the Figure 1 presented in [1].

[1] https://arxiv.org/pdf/2202.11783.pdf (TPAMI 2023)

---

> ### Author Response · Authors · 2023-11-20
> **Response to Reviewer GZ1m (1/3)**
>
> Thank you so much for your insightful reviews and constructive feedback on our manuscript. Based on your comments we have revised our manuscript. Additionally, below we further address the concerns and questions you raised.
>
> **Originality**
>
> We acknowledge your concerns about the originality of our contributions. This manuscript notably advances previous work by integrating innovative fairness-enhancing elements into the ARMED framework. Here are some key aspects of our extension:
>
> * The initial analysis of the ARMED model did not explicitly address fairness. Upon implementation, we discovered that while ARMED ME exhibits marginally better fairness than ARMED FE, the introduction of debiasing heads significantly enhances fairness in both outcomes. This was an unexpected finding, demonstrating the nuanced relationship between the model's components and fairness.
>
> * Our exploration included both Absolute Correlation Loss (ACL) and debiasing heads. Through comprehensive testing, we found debiasing heads to be consistently more effective. This discovery is pivotal, as it showcases that our framework can efficiently and simultaneously address both fairness and mitigate cluster confounding effects—a significant step forward in machine learning models.
>
> * Implementing this fairness enhancing MEDL framework was far from straightforward, due to the complexity of balancing multiple losses (MixedEffect, FixedEffect, various adversarial losses, and the Bayesian neural network loss). It required extensive trial and error to develop an effective method for hyperparameter optimization, ensuring stable operation of the framework. This process itself was a substantial undertaking, underscoring the novel challenges and solutions that our work presents.
>
> Overall, our advancements are not merely iterative but represent a substantial contribution and progression in developing more equitable and confounder-robust machine learning models.
>
> **Clarity**
>
> Regarding figure1. We understand your point. We felt the ARMED authors figure was well done, so while we drew our own figure from scratch, we styled it after the ARMED authors. We apologize if it appeared to be a cut and paste. We have updated the manuscript to use a different style for the figure so that it is clearly shown to be our own, new figure. The new figure removed undefined terms such as "h", "𝛽", and "m".  The new figure is clearer and more easy to understand.
>
> Based on your feedback, we have revised our Introduction to eliminate superfluous discussions and unnecessary literature references. We streamlined the content to focus more directly on the essence of our work, thereby enhancing clarity and reducing potential confusion for our readers.

---

> > ### Author Response · Authors · 2023-11-20
> > **Response to Reviewer GZ1m (2/3)**
> >
> > **Quality**
> >
> > Our choice of metrics was discussed in section 2.1; TPR and FPR std dev are conventionally used to assess equality of odds fairness in binary classification tasks [1]. By definition, equality of odds for binary classification takes the form: P(Ŷ=1 | A=0, Y=y) = P(Ŷ=1 | A=1, Y=y), y ∈ {0,1} where Ŷ is the predicted outcome, y is the ground truth outcome, and A is the sensitive subgroup membership. When y=1, this expression assesses true positive rates; likewise, y=0 is the expression of equalized false positive rates. The use of standard deviation of these rates for each sensitive subclass enabled us to recover a quantitative fairness “score” as described in a similar approach used by Yang et al. [2].
> >
> > To provide a more robust validation of our findings, we conducted additional experiments to provide additional baselines including the DFNN and DFNN adv. deb. These additional results appear in Tables 6 and 7. These results show that the proposed fair MEDL framework alone provides both increased fairness (e.g., over DFNN baseline) and improved accuracy (e.g., over the DFNN adv. deb.). This underscores the need for the proposed fair MEDL framework with both FixedEffects and RandomEffects (RE) subnetworks.
> >
> > Furthermore, we also added results for the ARMED model incorporating Absolute Correlation Loss (ACL) in Tables 8 and 9. These findings highlight a mixed impact on fairness enhancement and a substantial decrease in predictive accuracy, reaching up to a 6% reduction. This evidence further supports our choice of using debiasing heads, which consistently enhance fairness with only a marginal impact on predictive accuracy, generally under 1%.
> >
> > Thank you very much for your advice; we added Figure 4, 5 and 6 to visualize the trade-off between accuracy and fairness enhancement. We observed substantial improvements in both TPR SD and FPR SD, with fairness enhancements reaching up to 67% at a minimal accuracy trade-off of about 1%. Notably, the MixedEffect (ME) models outperform their FixedEffect (FE) counterparts in both accuracy and fairness, underscoring the efficacy of MixedEffect models in balancing fairness with accuracy. We recognize that the larger standard deviations in fairness and accuracy metrics for ARMED models could be attributed to their complexity, resulting in broader convergence areas post-training.
> >
> > **Significant**
> >
> > We are grateful for your suggestion to underscore the broader significance of our work. In response, we have included a new paragraph in the Discussion section to emphasize the real-world implications of our fairness improvements. This paragraph highlights how our framework can lead to more equitable decision-making in essential areas such as financial services and healthcare, directly benefiting everyday life. The specific paragraph is as follows:
> >
> > “Our research demonstrates substantial improvements in fairness, with direct implications for everyday applications. The Adult and IPUMS datasets are illustrative of pivotal income predictions that machine learning models make, which play a role in critical financial decisions such as loan and credit approvals. By applying our framework, these processes can become significantly fairer, ensuring equitable treatment across diverse demographics. Similarly, the Heritage Health dataset focuses on predicting future hospital utilization for individuals, which plays a central role in the calculation of insurance premiums, typically a major expense. Fairness in these predictions is crucial, as it directly impacts individuals' financial obligations for healthcare. Thus, our framework not only enhances model fairness but also contributes to more equitable decision-making in crucial sectors affecting daily life.”
> >
> > Additionally, we have revised our figures to better highlight the advancements made in this paper compared to the older ARMED model. We believe these updates more effectively showcase the unique contributions and value of our current work."
> >
> >
> > [1] Moritz Hardt, Eric Price, and Nathan Srebro. Equality of opportunity in supervised learning. In 30th Conference on Neural Information Processing Systems (NIPS 2016), 2016.
> > [2] Yang, Jenny, et al. "An adversarial training framework for mitigating algorithmic biases in clinical machine learning." NPJ Digital Medicine 6.1 (2023): 55.

---

> ### Author Response · Authors · 2023-11-20
> **Response to Reviewer GZ1m (3/3)**
>
> **Answers to your questions**
>
> 1. Our enhancements to the ARMED model significantly extend its capabilities. We expanded its application to regression tasks, diversifying its usability. Our key innovation was exploring two fairness-enhancing methods: Absolute Correlation Loss and debiasing heads. The integration of debiasing heads proved more effective, consistently improving fairness with negligible impact on accuracy. The major challenge was efficiently training a model with multiple submodels (FE, RE, cluster adversarial, and two debiasing heads), each requiring unique hyperparameters. We employed Bayesian Optimization and Hyperband (BOHB) for effective hyperparameter tuning. This approach balanced the complex interactions of multiple components, ensuring stable model performance. This development not only broadens the scope of ARMED but also adds technical depth, differentiating it substantially from the original model.
>
> 2. We refined all of our figures to make them more clear. We believe the updated figures clearly highlight the substantial contributions of our work.
>
> 3. Our choice of metrics was discussed in section 2.1. As further stated above, we utilized TPR and FPR std dev to assess equalized odds in our binary classification datasets. We chose equalized odds as our fairness metric for several reasons, including its inherent connection to model prediction accuracy and its applicability to both classification and regression tasks. For regression tasks (e.g. Heritage Health future hospital utilization rate), equalized odds fairness is defined through the standard deviation of the mean squared error (MSE) of the regression accuracy.
>
> 4. Thank you very much for your suggestions, it is an excellent way to visualize the trade-off between accuracy and fairness. We added figure 4, 5 and 6 following your advice. Figure 4 and 5 show substantial improvement on fairness while figure 6 shows negligible difference in accuracy across models.
>
>  5. and 6. Thank you so much for your suggestion. We added one paragraph discussing real-world applications in the Discussion section.

---

> ### Comment · Reviewer_GZ1m · 2023-11-21
> **Rating and Feedback: A Path to Improvement**
>
> As you may have observed, the overall rating falls below the average. I trust that the provided comments will assist you in revising the manuscript and enhancing its suitability for publication.

---

### Official Review · Reviewer_Nt9q · 2023-11-02

**Soundness:** 2 fair
**Presentation:** 2 fair
**Contribution:** 1 poor
**Rating:** 3
**Confidence:** 5

**Summary:**

This   paper   introduces   a   novel   Fairness-Enhancing   Mixed   Effects   Deep   Learning (MEDL)   framework   that   addresses   two primary   issues   in   traditional   deep   learning (DL):   the   failure   to   account   for   non-independent   and   identically   distributed   (iid) training   samples   in   clustered   data,   and   biases   toward   the   majority   group   in   the training   data,   which   can   have significant   repercussions   in   fields   like   finance   and healthcare.   The   framework   aims   to   enhance   fairness   while   maintaining   prediction performance and interpretability.
The   authors   propose   the   full   fairness-enhancing   ARMED   framework,   which   adds additional   adversarial   debiasing   subnetwork   to   the   original  ARMED   framework   for fairness-promoting.   Authors   claim   that   the   combination   of   ARMED   and   domain adversarial   debiasing   method   significantly   boosts   the   fairness   of   the   model,   and shows   the   consistent
improvements   of   the   new   framework   across   three   distinct datasets adopted in this study

**Strengths:**

1. The proposed framework is commendable for its innovative integration of cluster adversary, Bayesian neural networks, and a mixing function, enabling the distinction between cluster-invariant fixed effects (FE) and cluster-specific random effects (RE).
2.   The   robust   empirical   testing   across   diverse   datasets   (census/finance   and healthcare) and task types (classification and regression) validates the framework's applicability and effectiveness.
3.  The   significant   improvements   in   fairness   (up   to   86%   in   some   variables)   without substantial   loss   in   accuracy   is   a   remarkable   achievement,   showcasing   the framework's potential to balance fairness and performance effectively.

**Weaknesses:**

1. The Results section of the paper mainly explains the improvement of fairness of the   new   framework.   There   is   a   limited   explanation   of   “preserve   interpretability advantages” in the paper, which is mentioned in the abstract.
2. In section 3.1, the authors state that “while both Domain Adversarial Debiased (fair DA   adv.   deb.)   and   Domain   Adversarial   with   absolute   correlation   (fair   DA   ACL) enhance fairness, fair DA adv. deb. exhibits a more consistent fairness improvement across all sensitive variables.” Given the results in Table 1, it is difficult to see that fair DA  adv.  deb.  exhibits  a  more  consistent  fairness  improvement.  For  many  sensitive variables, the TPR or FPR standard deviation of fair DA ACL is smaller than fair DA adv.   deb..   Especially   for   the   Marital-status   feature   on   occupations   seen   during training and Sex feature on occupations unseen during training, fair DA ACL has both TPR   and   FPR   standard   deviation   smaller   than   fair   DA  adv.   deb.,   which   indicates
better fairness according to the paper. Authors then state that “Moreover, fair DA adv. deb. enhances fairness with minimal
reduction   in   balanced   accuracy   compared   with  fair   DA ACL—1%   vs   1.6%.   Given these   findings,   we   chose   to   incorporate   fair   DA   adv.   deb.   into   the   ARMED framework”.   The   balanced   accuracy   between   fair   DA ACL   and   fair   DA  adv.   deb. seems to be quite small and does not give a convincing reason for choosing fair DA adv. deb. over fair DA ACL
3: I would like to congratulate the authors on publishing their paper on TPAMI, however, ICLR is a decent venue for machine learning too. Advertising ARMED (Sec 2.2) can hardly be part of "Methods" as it is not an innovation/contribution in this paper. Note that unlike many conferences in signal processing, conference papers in top ML venues are not shortened versions of journal papers. You need to be substantially different and innovative from earlier works. Besides, the purpose of double blind is to remove the selection bias in favor of big names. As a reviewer for ICLR, knowing you have a TPAMI paper will not really affect my rating.

**Questions:**

There are too few baselines to conclude that the experiments are comprehensive enough to demonstrate its fairness.
The   authors   incorporate   fair   DA  adv.   deb.   instead   of   fair   DA ACL  into   the  ARMED framework based on their findings of the ablation study in section 3.1. But the reason for choosing fair DA adv. deb. over fair DA ACL is somehow not very convincing.
Authors can try to use the absolute correlation loss mentioned in fair DA ACL for the full fairness-enhancing ARMED framework, then make a comparison with the original proposed full fairness-enhancing ARMED.

**Details Of Ethics Concerns:**

This is arguably a dual submission - there is only incremental changes in this paper from their earlier TPAMI paper.
The authors used a significant portion highlighting their previous publication.

---

> ### Author Response · Authors · 2023-11-20
> **Response to Reviewer Nt9q (1/2)**
>
> Thank you for your valuable and constructive feedback. We sincerely appreciate the time and effort you have invested in reviewing our manuscript. Below, we address each of the questions you raised.
>
> **Limited Explanation of “Preserve Interpretability and Advantages”**
>
> We acknowledge the observation regarding the limited discussion on preserving interpretability advantages. What we mean in our approach to make MEDL fairer, we preserve the original MEDL framework’s ability to learn cluster specific effects and cluster-agnostic effects. For example, in this work for the ADULT dataset the MEDL model learns occupation independent effects through the FE subnetwork. Meanwhile, the RE subnetwork learns occupation specific effects. The way in which we make the MEDL framework more fair preserves this ability, and yet makes each subnetwork more fair. Having the ability to query our fair MEDL framework for both the learned fair cluster specific effects and learned fair cluster agnostic effects, provides more insight into what the model has learned than a conventional neural network which does not differentiate cluster specific from cluster agnostic effects. In this way our proposed fair MEDL framework is more interpretable.  We have incorporated a detailed explanation supporting this claim in a newly added paragraph within the Discussion section on page 9
>
> **Fairness Improvement Comparison (fair DA adv. deb. vs. fair DA ACL)**
>
> You correctly noted the discrepancies in our comparison of fairness improvements between the fair DA adv. deb. and fair DA ACL models. We realize that our explanation might have lacked sufficient clarity. Table 5 in the Appendix shows that the drop in ACL is 1.5% in unseen occupations. We also extended ARMED with ACL and include those results in table 8 and 9 in the Appendix. These results  demonstrate the impact of implementing Absolute Correlation Loss (ACL) in the ARMED framework as an alternative to the debiasing heads. While ARMED ACL shows variability in fairness improvement across different sensitive features, it also leads to a significant decrease in prediction accuracy, reaching up to a 6% reduction. Considering these outcomes, we assert that adding debiasing heads, with its more uniform improvement in fairness and negligible impact on accuracy, is a preferable option compared to ACL, at least across all 3 datasets we evaluated in this work.
>
> **Concern Regarding Dual Submission**
>  We understand your concern regarding the similarity of the figure we drew from scratch to that in the TPAMI paper. We have redrawn this figure to make it clear that this work is independent of any prior work. Additionally, this current manuscript is a significant contribution by investigating how fairness-enhancing components could be added to a MEDL framework, and many of these results will apply regardless of the specific implementation of MEDL (ARMED or domain adversarial) as we show in this manuscript. Below we list our specific contributions:
>
> It was non-obvious that MEDL frameworks, such as but not limited to ARMED, are unfair. They (e.g. ARMED, and domain adversarial) achieve many benefits, it could be the case that they are already fair. This work demonstrates that despite their other benefits in correcting bias due to cluster confounding, MEDL frameworks have unfair. We document and quantify the base level of unfairness. Additionally, we discover that the ME predictions are somewhat more fair than the FE but that adding the debiasing heads greatly improves fairness of both types of predictions. None of these findings were obvious outcomes, that this manuscript produces.
>
> We tried both ACL and the debiasing heads methodologies and found that the debiasing heads consistently work better. Our works demonstrate that fairness and batch effect mitigation can both be handled simultaneously and efficiently. This is a fundamental achievement that many machine learning researchers and projects could benefit by.
>
> We faced many challenges implementing the framework; i.e. its implementation is non-trivial. As an example, consider that there are multiple loss terms (MixedEffect fidelity, FixedEffect fidelity, three different adversarial losses, the Bayesian neural network KL-divergence loss). We document how these hyperparameters can be optimized to improve fairness while mitigating the confounding from sample clustering.

---

> > ### Author Response · Authors · 2023-11-20
> > **Response to Reviewer Nt9q (2/2)**
> >
> > **Suggestion for Additional Baselines:**
> >  We appreciate your suggestion to incorporate absolute correlation loss as used in fair DA ACL into the full fairness-enhancing ARMED framework for a more comprehensive comparison. These results are in table 8 and 9 in Appendix.
> >
> > We also implemented Yang’s method as additional baseline. These results are in table 6 and 7. The results are similar to the FixedEffects subnetwork, with strengthens our belief that making a full MEDL framework fair (with its added RandomEffects subnetwork) is preferable as it improve the model predictive accuracy and handles the confounding simultaneously, compared to Yang’s approach.

---

### Official Review · Reviewer_vJ3t · 2023-11-07

**Soundness:** 2 fair
**Presentation:** 3 good
**Contribution:** 1 poor
**Rating:** 3
**Confidence:** 4

**Summary:**

The authors build on prior work defining mixed effects to provide fair and robust predictors (ARMED). They add a debiasing term to ensure fairness in the fixed-effect part and demonstrate how their method improves on the ARMED baseline in terms of fairness using 3 datasets.

**Strengths:**

**Originality**:  the authors propose amending the ARMED framework to include a fairness regularizer. The combination of mixed effects (as defined by the authors) and fairness is novel.

**Quality**: the authors include multiple datasets and investigate many different attributes, not restricting themselves to binary classification and binary attributes.

**Clarity**: overall the paper is clear, although the RE part of the network could be explained a bit more.

**Weaknesses:**

**Originality**: this feels like a minor modification of the ARMED framework, especially as the $L_{CCE}(S,S')$ is the same as the loss on Z.

**Quality**: I believe important baselines are missing, as well as a proper discussion. For instance, the `fairness under distribution shift' is an important related field that is not cited here. Baselines from this field could be implemented, including some that include adversarial losses [1]. From my understanding, there isn't any baseline implemented outside of ARMED, even though Yang et al., 2023 is referenced.

In terms of the motivation of the method, I have major concerns:
- the fairness loss is on the FE part of the network. What prevents the RE part of the network from inducing bias? This is actually suggested by the better fairness results from the FE network compared to its ME counterpart.
- the authors mention that they enforce ‘equality of odds’, but they actually enforce that the model is not able to ‘encode’ the sensitive attribute. These are different criteria, and it is possible for models to encode a signal at the same level but display very different equalized odds [2].

**Significance**: the results seem quite variable (looking at the 95% CI), with obvious overlaps between multiple methods especially when considering unseen clusters. Can the authors discuss the additional complexity of ARMED compared to its variance, and benefit? It would also be good to mention how statistical significance is established (which test, n, and correction for multiple comparison). In addition, please see my question on optimization below.

[1] Schumann et al., 2019. Transfer of machine learning fairness across domains.
[2] Brown et al., 2023. Detecting shortcut learning for fair medical AI using shortcut testing.

**Questions:**

Scalability: the authors investigate different numbers of samples, but the number of features remain small (max 19). Can the authors comment on the implications of e.g. using images?

Optimization: the loss term includes multiple adversaries, mixes of loss types (e.g. cross-entropy with MSE), each with their own parameter. This seems like a difficult function to optimize, as even one term with an adversary can be challenging to converge. Can the authors comment on this?

---

> ### Author Response · Authors · 2023-11-20
> **Response to Reviewer vJ3t (1/2)**
>
> Thank you for your insightful review and valuable feedback on our manuscript. We appreciate your recognition of our work's originality and quality. Based on your comments we have revised our manuscript. Additionally, below we further address the questions you raised.
> **Originality and Quality:**
> While we concur that our work builds upon the ARMED framework,  we respectfully disagree that it is “a minor modification of the ARMED framework”. We have verified that MEDL does NOT intrinsically address fairness. We have quantified that inherent unfairness. We assert that the incorporation on fairness regularizers is a considerable advancement rather than a minor extension. Our exploration of both absolute correlation loss (ACL) and adversarial debiasing heads revealed that debiasing heads are consistently more effective at achieving fairness in a mixed effects context. This finding is a noteworthy discovery and is not an obvious result. The contribution we make and underscores our framework's ability to simultaneously and efficiently address fairness and batch effects. Notably, our approach integrates two debiasing heads, targeting the outputs of both FixedEffect and MixedEffect components. The implementation complexity, involving multiple losses (MixedEffect, FixedEffect, three adversarial losses, and the Bayesian neural network loss), necessitated extensive experimentation and refinement in hyperparameter optimization to achieve stable operation of our enhanced framework.
>
> In response to your concern regarding missing baselines, we have included results using Yang’s method in table 6 and 7. The results show that it performs similarly to our FixedEffect subnetwork's outcomes. This similarity bolsters our argument that the addition of the RandomEffect subnetwork marks a substantial improvement. By effectively managing batch effects, our model demonstrates enhanced capacity for fairness enhancement, as indicated in our revised results.
>
> We also added ARMED ACL as another baseline in table 8 and 9. These baselines show a mix in fairness enhancement, but the drop in predictive accuracy is much bigger – up to 6%. Based on these findings, we are confident that adding debiasing adversary networks is a better way to enhance fairness than using ACL, in the context of mixed effects deep learning.
>
> **Concerns about fairness Loss and enforcing equalized odds**
>
> We value your insight regarding the fairness loss being applied solely to the FE part of the network. To address potential bias in the RE component, we conducted additional experiments with dual debiasing heads: one for the FE and one for the RE portion of the mixed effects deep learning (MEDL) framework. The outcomes, presented in updated Tables 2 and 3, exhibit consistent fairness enhancement across both FE and ME outputs. Notably, ME displays superior fairness in seen occupations, both in baseline and fair implementations. However, in unseen occupations, we observed some variability in fairness improvement, suggesting the need for enhancements in our current z-predictor. This aspect is a key focus for our future research.
>
> Regarding the enforcement of equalized odds, we respectfully disagree with the statement that what we "enforce is that the model is not able to ‘encode’ the sensitive attribute.”  The cluster adversary for the FE subnetwork encourages that the conventional model does not encode cluster specific attributes. However, the debiasing heads, are structured quite differently from the cluster adversary. These debiasing heads only take the estimated prediction, y_hat and the ground truth prediction y as input and penalize the corresponding MEDL subnetwork for making errors in a biased way (e.g., more errors for females than males, or for one race versus another). The debiasing heads do not take the intermediate layers or features into consideration, only the errors made in prediction.
>
> Also, we deliberately chose equalized odds as the fairness approach, over other approaches such as demographic parity because 1) equalized odds was developed to overcome limitations in earlier methods such as demographic parity. 2) Equalized odds is applicable to both classification and regression tasks, 3) Equalized odds can be implemented readily as a neural network, facilitating integration to other deep learning frameworks, such as MEDL. 4) Equalized odds can effectively handle sensitive features encompassing more than two categories. These advantages align well with our framework, which utilizes neural networks for fairness improvements in both classification and regression, often dealing with multi-category sensitive features. Furthermore, our empirical results support the effectiveness of this approach in enhancing fairness across the datasets we examined.

---

> ### Author Response · Authors · 2023-11-20
> **Response to Reviewer vJ3t (2/2)**
>
> **Variability of Results in unseen clusters.**
> We recognize the variability in our results, particularly with unseen clusters. This variability may stem from the current limitations of our z-predictor, especially when faced with complex site invariance. We are actively working on enhancing our z-predictor by refining hyperparameter optimization and increasing its complexity to better handle such scenarios. These improvements are slated for future development and have been duly noted in the Discussion section of our manuscript.
>
> **Scalability and Optimization.**
> The complexity of the debiasing heads (AF and AM) is not a function of the number of features. Their complexity is fixed with respect to the number of features. Thus our approach does not increase in complexity with the number of features and should scale well with increasing feature count.
>
> What does increase in complexity with the number of features is the cluster adversary, Ac, and the conventional model. However, in the original MEDL framework that we chose to use here, (ARMED), those authors showed that their model does scale to images and video data.
>
> On the optimization front, we value your recognition of the inherent challenges. Achieving stable operation of our framework involved extensive experimentation. We ultimately employed BOHB [1] for simultaneous optimization of all weight_loss parameters associated with different loss terms. Additionally, we leveraged the Ray [2] framework for efficient parallel simulations. For instance, generating results for Table 2, where approximately 14,000 models were trained in parallel on 16 NVIDIA V100 GPU cards, was accomplished in just a few hours, demonstrating the framework's efficacy and efficiency.
>
> [1] Stefan Falkner, Aaron Klein, and Frank Hutter. BOHB: Robust and efficient hyperparameter optimization at scale. In International Conference on Machine Learning, pp. 1437–1446. Proceedings of Machine Learning Research, 2018.
>
> [2] Moritz, Philipp, et al. "Ray: A distributed framework for emerging {AI} applications." 13th USENIX Symposium on Operating Systems Design and Implementation (OSDI 18). 2018.

---

> ### Comment · Reviewer_vJ3t · 2023-11-21
> **Responses**
>
> I thank the authors for taking my feedback into account.
>
> However, my main concerns remain as it is unclear to me how this method compares to other (adversarial) techniques that do not use mixed effects.
>
> My suggestions are to include these baselines, but also to really think how their architecture addresses fairness. For instance, using the adversary in the FE or RE components makes different fairness assumptions, and might privilege best model per subgroup versus best model across groups. These are interesting directions to investigate and would add insights and novelty to the paper, compared to the extension that is currently proposed.

---

### Meta-Review · Area_Chair_jyCZ · 2023-12-10

**Metareview:**

As discussed between the reviewers , SAC and myself, the paper breaks the anonymity, which is enough ground for rejection. In addition, the reviewers pointed out several issues (especially with regards to the novelty of the work) which I encourage the authors to  address in the revised version of their paper.

**Justification For Why Not Higher Score:**

Violation of double-blindness and several major concerns, especially with regards to novelty.

**Justification For Why Not Lower Score:**

N/A

---

### Decision · Program_Chairs · 2024-01-16

Reject